# Germ granule compartments coordinate specialized small RNA production

Xiangyang Chen[1,6], Ke Wang[1,6], Farees Ud Din Mufti ®[1,6], Demin Xu[1], Chengming Zhu ®[1], Xinya Huang[1], Chenming Zeng[1], Qile Jin[1], Xiaona Huang[1], Yong-hong Yan[2], Meng-qiu Dong ®[2], Xuezhu Feng ®[3] ✉, Yunyu Shi ®[1] ✉, Scott Kennedy ®[4] ✉ & Shouhong Guang ®[1,5] ✉

Germ granules are biomolecular condensates present in most animal germ cells. One function of germ granules is to help maintain germ cell totipotency by organizing mRNA regulatory machinery, including small RNA-based gene regulatory pathways. The *C. elegans* germ granule is compartmentalized into multiple subcompartments whose biological functions are largely unknown. Here, we identify an uncharted subcompartment of the *C. elegans* germ granule, which we term the E granule. The E granule is nonrandomly positioned within the germ granule. We identify five proteins that localize to the E granule, including the RNA-dependent RNA polymerase (RdRP) EGO-1, the Dicer-related helicase DRH-3, the Tudor domain-containing protein EKL-1, and two intrinsically disordered proteins, EGC-1 and ELLI-1. Localization of EGO-1 to the E granule enables synthesis of a specialized class of 22G RNAs, which derive exclusively from 5′ regions of a subset of germline-expressed mRNAs. Defects in E granule assembly elicit disordered production of endogenous siRNAs, which disturbs fertility and the RNAi response. Our results define a distinct subcompartment of the *C. elegans* germ granule and suggest that one function of germ granule compartmentalization is to facilitate the localized production of specialized classes of small regulatory RNAs.

Biomolecular condensates, such as nucleoli, processing bodies, Cajal bodies, stress granules and germ granules, are micron or submicron scale compartments in cells, that lack surrounding membranes and concentrate specific molecules into distinct subcellular spaces[1–4]. The formation of many condensates is driven by liquid-liquid phase separation (LLPS), likely via multivalent interactions between RNA, intrinsically disordered proteins, and RNA-binding proteins[2,5]. These liquid-like condensates are structurally dynamic, with molecular constituents exchanging rapidly with the surrounding cytoplasm or nucleoplasm[3]. Current models posit that one function of biomolecular condensates is to spatiotemporally compartmentalize specific proteins and nucleic acids within distinct subcellular environments, thus providing cells with strategies and opportunities for organizing and regulating specific biochemical reactions and gene expression programs[2,6].

[1]Department of Obstetrics and Gynecology, The First Affiliated Hospital of USTC, The USTC RNA Institute, Ministry of Education Key Laboratory for Membraneless Organelles & Cellular Dynamics, Hefei National Research Center for Physical Sciences at the Microscale, Center for Advanced Interdisciplinary Science and Biomedicine of IHM, School of Life Sciences, Division of Life Sciences and Medicine, Biomedical Sciences and Health Laboratory of Anhui Province, University of Science and Technology of China, Hefei, Anhui 230027, China. [2]National Institute of Biological Sciences, Beijing 102206, China. [3]School of Basic Medicine, Anhui Medical University, Hefei, China. [4]Department of Genetics, Blavatnik Institute at Harvard Medical School, Boston, MA 02115, USA. [5]CAS Center for Excellence in Molecular Cell Science, Chinese Academy of Sciences, Hefei, Anhui 230027, China. [6]These authors contributed equally: Xiangyang Chen, Ke Wang, Farees Ud Din Mufti. ✉e-mail: fengxz@ustc.edu.cn; yyshi@ustc.edu.cn; kennedy@genetics.med.harvard.edu; sguang@ustc.edu.cn

Germ granules are RNA-rich membrane-less condensates that are often found docked on the cytoplasmic surface of germline nuclei. Germ granules are present in the germ cells of many, if not all, animals including worms, flies, and mice[7,8]. Germ granules are thought to act as organizational hubs for posttranscriptional gene regulation to promote germ cell function[8]. In *C. elegans*, recent studies have found that the germ granule is compartmentalized into four seemingly distinct regions that encapsulate distinct sets of proteins in adults' germlines. One well-studied compartment of the germ granules is the P granule, which forms via LLPS and exhibits liquid-like behaviors[9,10]. P granules are thought to be major sites of mRNA regulation in germ cells, which serve as hubs for self/nonself RNA discrimination via small interfering (si) RNAs and the RNA interference (RNAi) machinery[11–15]. In addition to P granules, studies have identified three additional subcompartments of *C. elegans* germ granules in adult germlines: the Mutator foci, the Z granules, and the SIMR foci[6,16,17]. For simplicity's sake, P granules, Z granules, Mutator foci, and SIMR foci will also be referred to as the P, Z, M, and S compartments of the germ granule when mentioned simultaneously in the following context. Distinct proteins have been identified that mark the P, Z, M, and S compartments of the germ granule. For instance, PGL-1 is a germline-expressed protein that is a constitutive protein component of P granules[9,18,19]; MUT-16 is a low-complexity protein that marks Mutator foci[20,21]; ZNFX-1 is a conserved SF1 helicase domain-containing zinc finger protein, that marks Z granules[22,23]; and the Tudor domain protein SIMR-1 marks SIMR foci[24]. Little is known about how and why germ granules are divided into granular units or whether additional germ granule subcompartments await discovery.

The current model of germ granule organization in *C. elegans* posits that the subcompartments of the *C. elegans* germ granule are not randomly ordered with respect to each other within the larger germ granules. For instance, many germ granules contain a single Z granule sandwiched between a P granule and an Mutator focus, forming ordered tri-condensate assemblages termed PZM granules[22]. SIMR foci are also found in tripartite structures, adjacent to Z granules and opposite P granules, although the orientation of all four condensates relative to one another is still undetermined[24]. Interestingly, the architecture of germ granules varies during development. For instance, in the germline progenitor cells of early embryos, the Z granule proteins, ZNFX-1 and WAGO-4, colocalize to P granules rather than forming discrete structures; however, after the 100-cell stage of embryonic development, the Z granule demix into discrete condensates, adjacent to the P granule[22]. Similarly, during embryogenesis, the Mutator and SIMR foci factors diffuse evenly in the cytosol in early embryos, and they are first observed forming robust condensates around the 100-cell stage of embryonic development in Z2/Z3 progenitor germ cells[25]. A recent study identified a sperm-specific germ granule, termed the paternal epigenetic inheritance (PEI) granule, that mediates paternal epigenetic inheritance during spermatogenesis in *C. elegans*, which further suggests that the generation of compartments of germ granules is regulated by development stages[26]. Newly synthesized mRNAs and proteins may contribute to the dynamic architecture of germ granules[16]. However, it is unclear what is the driving force in germline condensate assembly in *C. elegans*.

Small RNAs termed siRNAs, piRNAs, and miRNAs, as well as the Argonaute (AGO) proteins that bind these small RNAs, are present in many eukaryotes where they regulate gene expression programs at both the transcriptional and posttranscriptional levels[27–31]. The *C. elegans* genome encodes 19 AGOs, which bind small RNAs that include miRNAs, piRNAs (also 21U RNAs), 26G RNAs and 22G RNAs[32]. A number of AGOs localize to the *C. elegans* germ granule[32]. For instance, the miRNA binding AGO ALG-5 and the 26G siRNA binding AGO ALG-3 and ALG-4 complexes localize to germ granules where they promote spermatogenesis[32–35]; the piRNA binding protein PRG-1 localizes to P granules, where it is thought to help germ cells identify and distinguish

self (germline mRNAs) from nonself RNAs (transposable elements)[36–39]. Additionally, several AGOs, which bind to 22G RNAs, localize to germ granular subcompartments. These include the P granule localized CSR-1 and WAGO-1, the Z granule localized WAGO-4 and PEI-granule localized PPW-2/WAGO-3[22,26,40,41]. A number of additional proteins that are implicated in the biogenesis or processing of small RNA are also reported to localize to germ granules. For instance, a number of factors involved in the maturation of piRNAs are thought to localize to the *C. elegans* germ granule. These include components of the PICS/PETISCO piRNA processing complex[42,43], PARN-1, which trims the 3′ ends of piRNAs[44], and HENN-1, which 2′-O-methylates piRNAs[45–47]. Additionally, proteins involved in 22G RNAs (22 nt long with 5′G) production are reported to localize to germ granules in germ cells[29]. 22G RNAs can be divided into two major classes: CSR-1-class 22G RNAs and WAGO-class 22G RNAs[29]. The current model posits that RdRP RRF-1 localizes to Mutator foci where it synthesizes WAGO-class 22G RNAs using pUGylated RNA fragments as templates[20,48]. CSR-1-class 22G RNAs are synthesized by RdRP EGO-1, perhaps in P granules[29,41]. Taken together, the results suggest that the germ granule is a major site for small RNA production and small RNA-based RNA surveillance in germ cells. Indeed, mutations that disrupt germ granule assembly are known to misregulate small RNA expression and disrupt gene expression programs, suggesting that germ granules are the major sites of small regulatory RNA-mediated gene regulation in the *C. elegans* germline[20,49–53]. How germ granules organize and regulate the complex small RNA pathways present in germ cells remains poorly understood.

Here we identify an uncharted subcompartment of the *C. elegans* germ granule, which we term the E granule. We identified five proteins that localize to the E granule, including the RNA-dependent RNA polymerase EGO-1, the Dicer-related helicase DRH-3, the Tudor domain-containing protein EKL-1, and two intrinsically disordered proteins, EGC-1 and ELLI-1. The E granule is nonrandomly positioned within the larger germ granule and is specialized for the synthesis of a specialized class of 22G RNAs, which derive exclusively from 5′ regions of a subset of CSR-1-class germline-expressed mRNAs. Defects in E granule assembly result in disordered synthesis of endogenous siRNAs, which disturbs fertility and the RNAi response. The data suggest that compartmentalization of the germ granule allows germ cells to produce distinct types of small RNAs, which greatly expands the diversity and regulatory potential of small RNA pathways in the germline.

## Results

### EGO-1 localizes to an unknown germ cell focus

EGO-1 and RRF-1 are RdRPs that synthesize siRNAs in the *C. elegans* germline using a largely nonoverlapping set of mRNAs as templates[29]. Because animals lacking EGO-1 are sterile, current models posit that siRNAs are essential regulators of gene expression in the *C. elegans* germline[54,55]. To better understand how and why small RNAs might be essential for germ cell function, we epitope-tagged the RRF-1 and EGO-1 proteins with the fluorescent proteins tagRFP and GFP, respectively (see methods). Epitope-tagged RRF-1 and EGO-1 encoded functional proteins, since the tagged animals exhibited similar feeding RNAi responsiveness and brood size to those of wild-type animals, respectively (Supplementary Fig. 1a–c). As previously reported, RRF-1::tagRFP accumulated in perinuclear foci surrounding germ cell nuclei and colocalized with MUT-16::GFP, which is a marker for Mutator foci (Supplementary Fig. 1d)[20].

EGO-1 is expressed in all embryonic cells in early embryos and is only expressed in Z2/Z3 germline cells in late embryos (Supplementary Fig. 2a). At the larval and adult stages, EGO-1 is expressed in the germline (Supplementary Fig. 2b, c). We observed that GFP::EGO-1 largely accumulated in perinuclear foci, which is consistent with a previous report[41], and also formed a considerable number of visible aggregates in the rachis of the germline (Fig. 1a and Supplementary

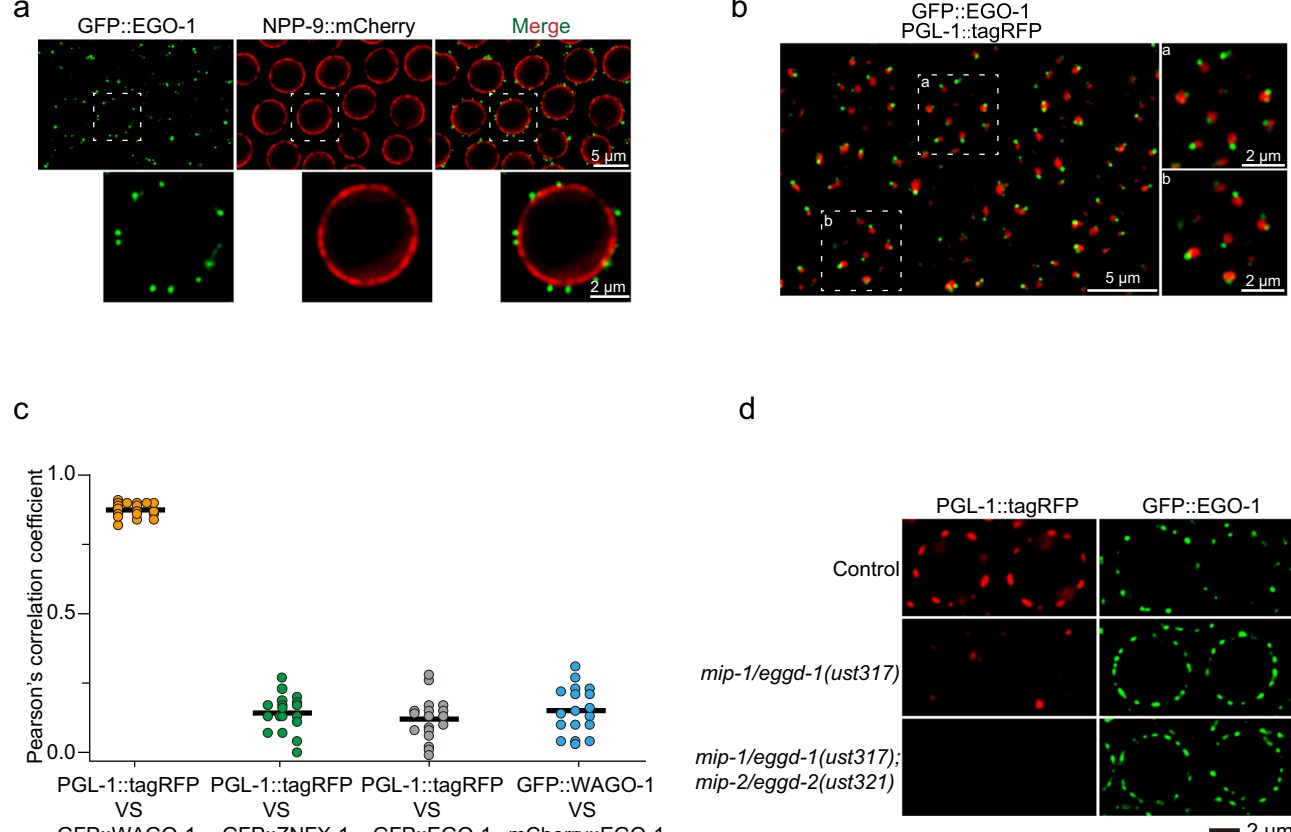

**Fig. 1 | EGO-1 localizes to unknown germ cell foci. a** Pachytene germ cells of animals that express GFP::EGO-1 and NPP-9::mCherry. NPP-9 is a putative homolog of UNP358 in humans, which forms part of the nuclear pore complex (NPC) cytoplasmic fibrils[11]. Fluorescence epitopes at the native chromosomal loci of each gene were tagged via CRISPR (see Methods). **b** Left. Fluorescence micrographs of pachytene germ cells that express GFP::EGO-1 and PGL-1::tagRFP. Right, representative pictures showing EGO-1 foci and PGL-1 foci. **c** Quantification of colocalization between the indicated fluorescent proteins in pachytene germ cells (see Methods). Each data point represents the Pearson's R value showing the degree of colocalization between two fluorescence channels covering an individual germ cell (18 germ cells in total from 3 animals). The mean is indicated by a solid black line. Source data are provided as a Source Data file. **d** Images of representative meiotic germ cells of the indicated animals. The loss of MIP-1/EGGD-1 or depletion of both MIP-1/EGGD-1 and MIP-2/EGGD-2 does not block the perinuclear localization of EGO-1. All images were taken by the Leica THUNDER imaging System and deconvoluted using Leica Application Suite X software (version 3.7.4.23463). All images are representative of more than three animals.

Fig. 2b–d). Recent works suggested that EGO-1 did not localize to Mutator foci and Z granules in the pachytene region of the germline[20,22]. We further observed that GFP::EGO-1 did not colocalize with SIMR-1::tagRFP, which is a marker for SIMR foci, and confirmed that GFP::EGO-1 did not colocalize with tagRFP::ZNFX-1 throughout the germline, which is a marker for Z granules (Supplementary Fig. 3a). Unexpectedly, we found that GFP::EGO-1 also did not colocalize with PGL-1::tagRFP, which is a commonly used marker for P granules (Fig. 1b and Supplementary Fig. 3b, c). Similarly, mCherry::EGO-1 did not colocalize with GFP::WAGO-1, which is another marker of P granules (Supplementary Fig. 3d)[40]. We quantified the overlap between GFP::EGO-1, PGL-1::tagRFP, and GFP::WAGO-1 fluorescence signals in animals expressing combinations of these fluorescent proteins (see Methods). The analysis showed that, as expected, PGL-1::tagRFP fluorescence overlapped extensively with GFP::WAGO-1 fluorescence (Fig. 1c). However, GFP::EGO-1 and mCherry::EGO-1 fluorescence did not overlap with PGL-1::tagRFP and GFP::WAGO-1 fluorescence respectively, suggesting that EGO-1 does not localize with PGL-1 or WAGO-1 in the P granule (Fig. 1c). Consistent with this idea, the depletion of MIP-1/EGGD-1 and MIP-2/EGGD-2, which are known to disrupt perinuclear P granule formation, did not affect the size or distribution of perinuclear GFP::EGO-1 foci (Fig. 1d)[56–58]. Together, we conclude that GFP::EGO-1 may not localize to known compartments of the germ granule and may accumulate to an unknown germ cell focus.

## Identification of EGO-1-interacting proteins that are required for the RNAi response

To further our understanding of EGO-1, small RNA-based gene regulation, and germ granule biology, we sought to identify EGO-1-interacting proteins using immunoprecipitation followed by mass spectrometry (IP-MS). The experiment identified a number of proteins that were enriched by GFP::EGO-1 immunoprecipitation (Fig. 2a). Two of these proteins, EKL-1 and DRH-3, have been reported previously to interact physically with EGO-1[40,59], indicating that our EGO-1 IP-MS was successful. Henceforth, the complex of EGO-1, DRH-3, and EKL-1 will be collectively referred to as the EGO module in the following context.

EGO-1, EKL-1 and DRH-3 are needed for the RNAi response by modulating the production of mRNA-templated short antisense RNAs[40,41,54,55]. We then tested whether other putative EGO-1 interactors are needed for feeding RNAi response. The top 12 IP-MS candidates were selected to be examined. We generated deletion alleles of these candidates (DRH-3 and EKL-1 were excluded) via dual sgRNA-directed CRISPR/Cas9 technology[60]. We successfully generated fertile nematode strains carrying putative null alleles of these genes, except for *vha-6* and *tni-1*, which may be essential genes needed for growth or fertility. A detailed description of these alleles, as well as the other mutant alleles used in this study, is listed (Supplementary Fig. 4a). We then examined whether these genes participate in exogenous RNAi processes by feeding animals bacteria expressing dsRNAs targeting nematode genes.

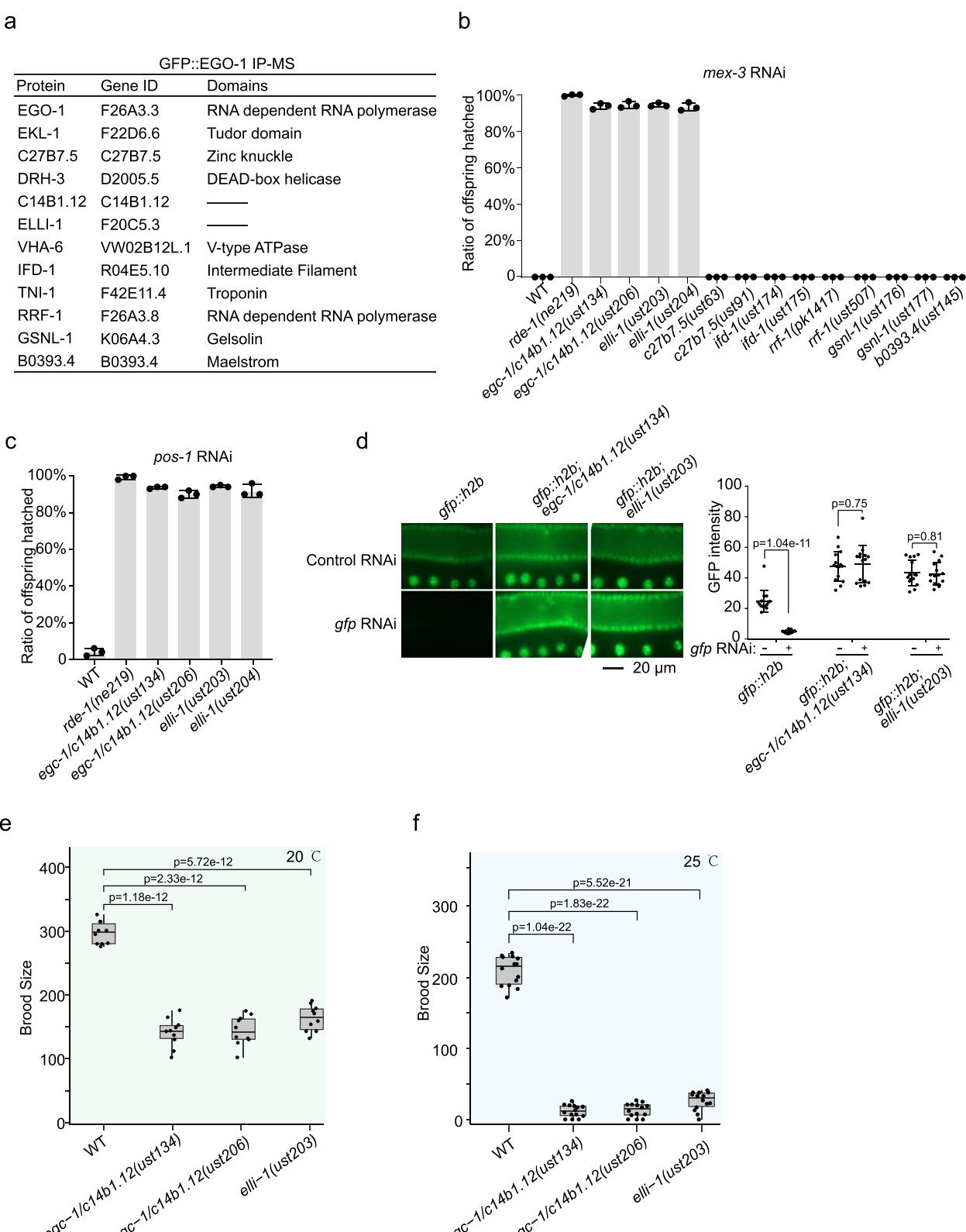

mex-3 encodes a KH domain protein that regulates blastomere identity in early *C. elegans* embryos, deletion of which causes unhatched embryos[61]. We fed the mutants with bacteria expressing dsRNAs targeting *mex-3* and found that, among the mutants, *elli-1(-)* and *c14b1.12(-)* animals were defective for experimental RNAi (Fig. 2b). ELLI-1 was previously identified to function with CSR-1 to modulate RNAi activity, P granule morphology, fertility and gene expression in the germline[62]. Based on the data described below, *c14b1.12* was named E granule component-1 (*egc-1*). Animals lacking EGC-1/C14B1.12 and ELLI-1 were also defective for experimental RNAi targeting *pos-1* (Fig. 2c). To further confirm the requirements of EGC-1/C14B1.12 and ELLI-1 in the germline RNAi response, we tested the silencing efficiency of a germline-expressed *gfp::h2b* transgene upon *gfp* RNAi. The loss of EGC-1/C14B1.12 or ELLI-1 significantly prohibited the silencing effect of

**Fig. 2 | Identification of EGO-1 interacting proteins that are required for feeding RNAi response. a** A list of selected top EGO-1 interacting partners identified by immunoprecipitation followed by mass spectrometry, based on WD value (IP-MS). A list of all proteins identified by GFP::EGO-1 IP-MS is shown in Supplementary data 1. Quantification of hatched embryos of the indicated animals after feeding RNAi targeting *mex-3* (**b**) and *pos-1* (**c**). *mex-3* and *pos-1* encode a KH domain protein and a zinc-finger protein respectively, that are required for early embryonic cell-fate decisions[61,89]. RNAi targeting *mex-3* or *pos-1* induces embryonic arrest in F1 embryos of animals exposed to dsRNA[89,90]. Synchronized animals of the indicated genotypes were cultured on plates seeded with bacteria expressing the indicated dsRNAs. The percentage of hatched embryos was scored. Data are presented as mean values +/- SD of three biologically independent samples. **d** EGC-1/C14B1.12 and ELLI-1 are required for feeding RNAi targeting germline expressed GFP. Animals expressing GFP::HIS-58 were exposed to *gfp* RNAi. Bleached embryos were cultured in RNAi plates seeded with bacteria expressing *gfp* dsRNA. Left: fluorescence images of the indicated animals without or with *gfp* RNAi. Right: GFP intensity levels of the indicated animals were measured by ImageJ. *N* = 3 biologically independent samples. Fifteen worms in total were measured for each animal. Data are presented as mean values +/- SD. A two-tailed *t*-test was performed to determine statistical significance. Brood sizes of the indicated animals at 20 °C (**e**) and 25 °C (**f**). Bleached embryos were hatched and grown at 20 °C or 25 °C. Then, L3 worms were transferred individually onto fresh NGM plates. The number of progeny worms was scored. *N* = 10 animals. Bolded midline indicates median value, box indicates the first and third quartiles, and whiskers represent the most extreme data points within 1.5 times the interquartile range. A two-tailed *t*-test was performed to determine statistical significance. Source data are provided as a Source Data file.

GFP::H2B upon feeding RNAi targeting *gfp* (Fig. 2d). However, the soma-expressed SUR-5::GFP transgene was still effectively silenced in *egc-1(-)* and *elli-1(-)* mutants, suggesting that EGC-1 and ELLI-1 are not needed for the somatic RNAi response (Supplementary Fig. 4b). It is noteworthy that the RNAi defects in the *elli-1* mutants generated for this study are much more pronounced than previously reported[62]. We speculate that the previous EMS mutagenesis-generated *elli-1* alleles are likely hypomorphic, which may not be null alleles and cause partial loss-of-function[62].

EGO-1 is needed for the fertility of *C. elegans*[41,55,63–65]. Similar to *ego-1* mutants and a previous report[62], animals lacking EGC-1 or ELLI-1 exhibited fertility defects; *egc-1(-)* and *elli-1(-)* animals produced 47.5% and 48.5% of the number of progeny as wild-type animals, respectively, when assayed at 20 °C (Fig. 2e); the number of *egc-1(-)* and *elli-1(-)* progeny dropped to 6.0% and 12.4% of wild-type animals, respectively, when assayed at 25 °C (Fig. 2f). Animals lacking both EGC-1 and ELLI-1 did not show further reduction of brood size, compared with that of *elli-1(-)* animals, suggesting that EGC-1 and ELLI-1 may regulate reproduction in the same pathway (Supplementary Fig. 4c).

### EGO-1 and its interacting partners localize to perinuclear foci in the germline

EGC-1 and ELLI-1 are exclusively needed for germline RNAi, suggesting that these two proteins may be exclusively expressed in the germline. Indeed, the *elli-1* mRNA has been reported to enrich in the germline[62]. To further investigate the expression patterns and subcellular localization of EGO-1 and EGO-1 interactors, we introduced GFP or mCherry fluorescence epitopes into the endogenous *egc-1, elli-1, drh-3* or *ekl-1* genes. All of these animals are responsive to feeding RNAi at similar levels to wild-type animals and produce normal numbers of progeny, suggesting that these modified genes encode functional proteins (Supplementary Fig. 5a, b). EGC-1::GFP, EKL-1::GFP and DRH-3::GFP were expressed in the germline and embryos; ELLI-1::GFP was expressed in the germline but was not detectable in embryos (Supplementary Fig. 5c–h). DRH-3 and EKL-1, but not EGO-1, EGC-1 and ELLI-1, were also expressed in somatic cells, which is consistent with their functions in the production of somatic siRNAs (Supplementary Fig. 5i)[40].

Adult animals possess two U-shaped gonad arms, in which multiple germ cells form a columnar monolayer (termed the surface) and are radially arranged around a central core of cytoplasm (termed the rachis), comprising a syncytial architecture[66]. We found that these proteins mainly accumulated on the surface of the adult germline, likely in perinuclear foci surrounding germ cell nuclei, and formed a considerable amount of smaller aggregates in the rachis (Fig. 3a and Supplementary Fig. 5j), which was consistent with the subcellular localization patterns of GFP::EGO-1 (Supplementary Fig. 2d) and ELLI-1::GFP[62]. Simultaneous imaging of these proteins and LMN-1, which marks the nuclear envelope, supported that these proteins mainly accumulated in perinuclear foci surrounding germ cell nuclei (Fig. 3b). We further examined the subcellular localization of ELLI-1 by imaging of two additional ectopically expressed ELLI-1 transgenes tagged with tagRFP or GFP, and found that the ectopically expressed ELLI-1::GFP*(ustIS272, LG II)* and tagRFP::ELLI-1*(ustIS268, LG I)* also mainly accumulated on the surface of the germline (Supplementary Fig. 6a). Additionally, simultaneous imaging of ELLI-1::GFP*(ust374, in situ)* and mCherry::CGH-1 both on the surface and in the rachis of the germline revealed that ELLI-1::GFP did not colocalize with mCherry::CGH-1, indicating that ELLI-1 does not accumulate in the P-body (Supplementary Fig. 6b).

Interestingly, EGO-1, DRH-3 and EKL-1 mainly diffuse throughout the cytosol during embryonic development, rather than accumulating in perinuclear foci (Supplementary Figs. 2a, 5h). In embryos, EGC-1 localized diffusely throughout both the cytoplasm and the nucleus without forming observable foci (Supplementary Fig. 5h). ELLI-1::GFP was barely detectable in early embryos, implying that the expression of ELLI-1 may be inhibited during early embryo development (Supplementary Fig. 5h). Consistently, the zygotic ELLI-1 was shown to begin to accumulate in the cytoplasm of primordial germ cells between the comma to 2-fold stage of embryogenesis[62]. To further examine the regulation of ELLI-1 expression in early embryos, we generated GFP, GFP::HIS-58, ELLI-1::GFP and tagRFP::ELLI-1 transgenes under the control of the *mex-5* promoter and *tbb-2 3'UTR* or *elli-1 3'UTR*. All of these transgenes expressed fluorescent proteins in the germline (Supplementary Fig. 6c). However, in early embryos, only GFP and GFP::HIS-58, but not ELLI-1::GFP and tagRFP::ELLI-1 could be detected, suggesting a possible posttranscriptional regulation of *elli-1* gene in early embryos (Supplementary Fig. 6c). These data suggested that the perinuclear localization of EGO-1 and EGO-1 interactors was dynamically regulated during development.

To assess if EGO-1 and its four interacting proteins might colocalize to the same germ cell foci, we used genetic crosses to generate animals expressing combinations of the GFP-, mCherry-, or tagRFP-tagged proteins described above. We found that EGC-1::GFP colocalized with mCherry::EGO-1, and GFP::EGO-1 colocalized with tagRFP::ELLI-1 (Fig. 3c). Image quantifications of the spatial overlap in fluorescent signals confirmed the above colocalization and established that tagRFP::ELLI-1 colocalized with EGC-1::GFP, DRH-3::GFP and EKL-1::GFP (Fig. 3d and Supplementary Fig. 7a). In addition, EGC-1::GFP, ELLI-1::GFP, DRH-3::GFP and EKL-1::GFP did not colocalize with a marker of P granules (PGL-1::tagRFP), suggesting that these proteins, similar to EGO-1, do not localize to P granules (Fig. 3d and Supplementary Fig. 7b–d). Interestingly, while the foci formed by these five proteins were distinct from P granules marked by PGL-1::tagRFP, these foci were almost always observed immediately adjacent to P granules (Supplementary Figs. 3b, d and 7b–d). Consistently, perinuclear ELLI-1 foci were shown to dock next to P granules in germ cells[62]. Overall, we conclude that EGO-1 and its interacting proteins mainly accumulate to unknown perinuclear germline foci, which form immediately adjacent to the P granules.

### EGO-1 and its interacting partners define the E granule

We wondered whether EGO-1 and its interacting proteins might localize to other known subcompartments or uncharted subcompartments of the germ granule. We first examined if EGO-1-interacting partners might

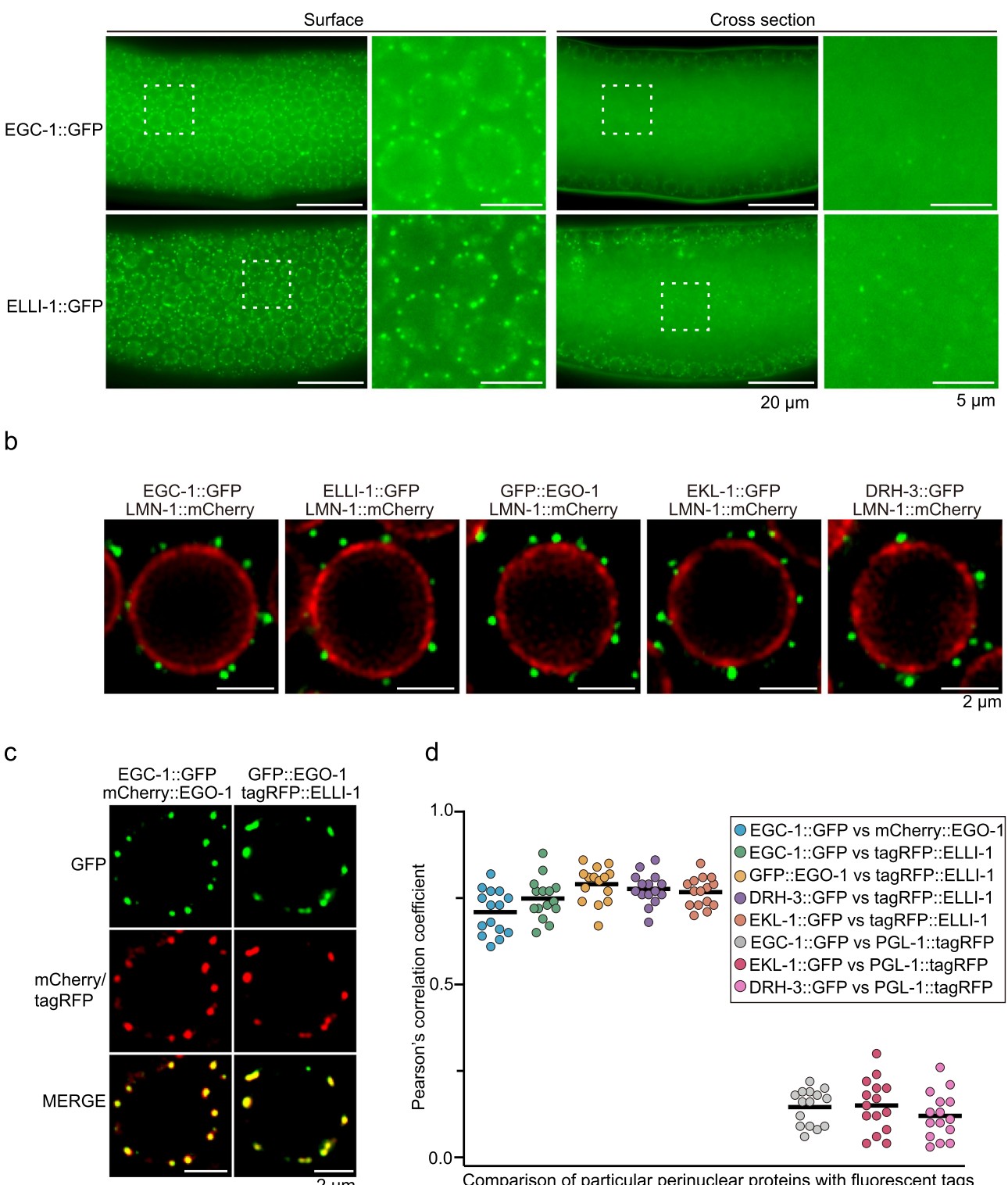

**Fig. 3 | EGO-1 and its interacting partners localize to perinuclear foci in the germline. a** Fluorescence micrographs of the surface and rachis of the germline in live adult animals expressing EGC-1::GFP or ELLI-1::GFP. **b** Fluorescence micrographs of pachytene germ cells that express LMN-1::mCherry and the indicated GFP-tagged proteins. **c** Pachytene germ cells of animals that express the indicated fluorescent proteins. **d** Quantification of colocalization between the indicated fluorescent proteins of pachytene germ cells (see Methods). Each data point represents the Pearson's R-value showing the degree of colocalization between two fluorescence channels covering an individual germ cell (15 germ cells in total from 3 independent animals). The solid black line indicates the mean value. All images were taken by the Leica THUNDER imaging System and deconvoluted using Leica Application Suite X software (version 3.7.4.23463). All images are representative of more than three animals. Source data are provided as a Source Data file.

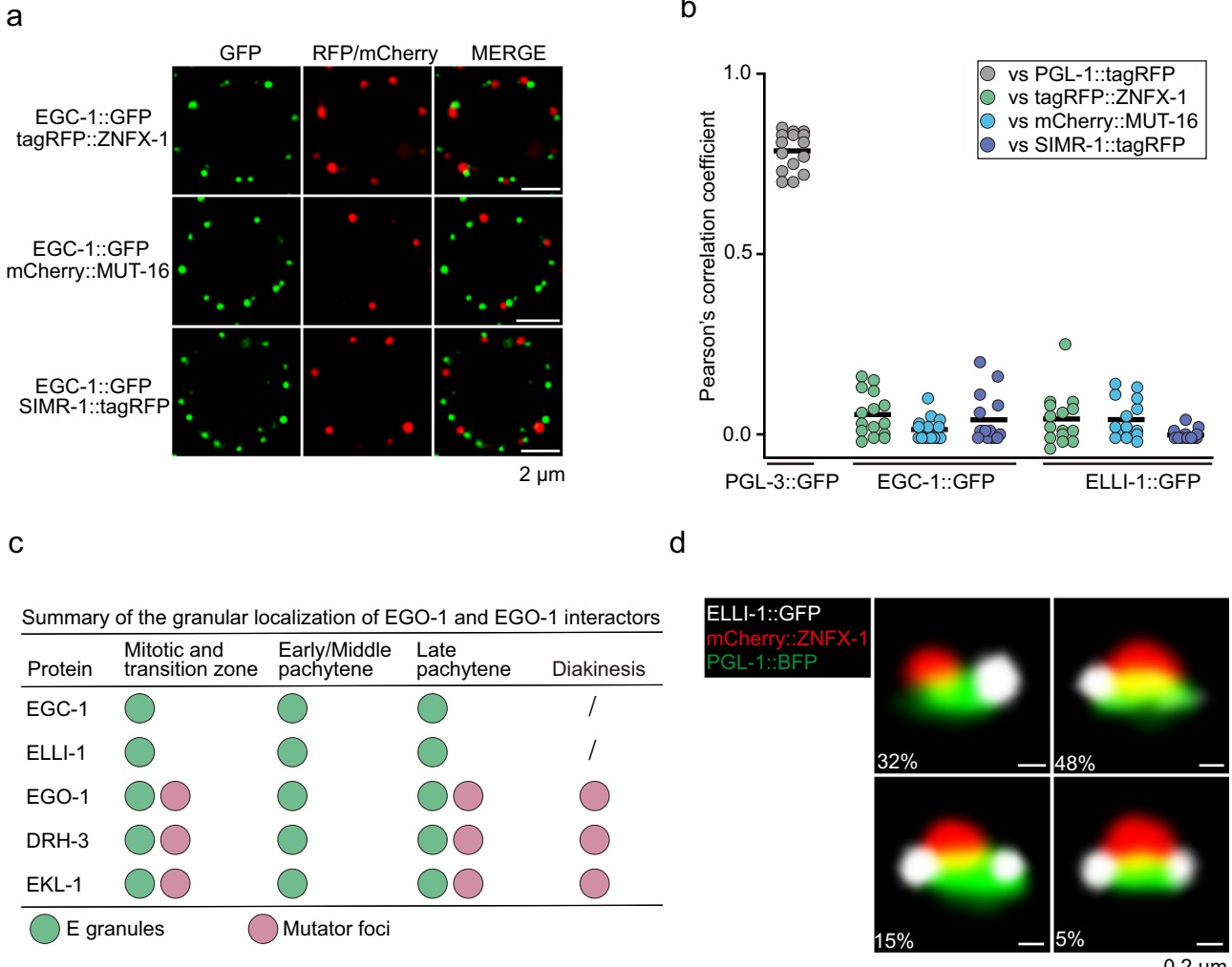

**Fig. 4 | EGO-1 and its interacting partners define the E granule. a** Pachytene germ cells of animals that express EGC-1::GFP and the indicated tagRFP- or mCherry- tagged proteins. All images are representative of more than three animals. **b** Quantification of colocalization between the indicated fluorescent proteins of pachytene germ cells. Each data point represents the Pearson's R-value showing the degree of colocalization between two fluorescence channels covering an individual germ cell (15 germ cells in total from 3 independent animals). The solid black line indicates the mean value. PGL-3 is a constitutive component of the P granule at all stages of development[19]. Source data are provided as a Source Data file. **c** Summary of the perinuclear localization of EGO-1 and EGO-1 interactors. Note that E granules disintegrates during diakinesis. The green and pink balls represent E granules and Mutator foci, respectively. **d** Representative pictures of germ granules in pachytene germ cells showing that the E granule is nonrandomly positioned within the germ granule. A total of 210 germ granules from 7 worms were analyzed. The percentage of sequential assembly of E, P and Z granules was counted. All images were taken by the Leica THUNDER imaging System and deconvoluted using Leica Application Suite X software (version 3.7.4.23463).

localize to the Z, S, or M compartments of the germ granule. EGC-1::GFP and ELLI-1::GFP formed foci that were distinct from the foci formed by tagRFP::ZNFX-1 (Z granules), mCherry::MUT-16 (Mutator foci), and SIMR-1::tagRFP (SIMR foci) throughout the germline (Fig. 4a and Supplementary Fig. 8a, b). Image quantification of spatial overlap between these fluorescence signals confirmed that EGC-1::GFP and ELLI-1::GFP localized to foci that were largely distinct from the Z, S, or M compartments of the germ granule (Fig. 4b and Supplementary Fig. 8c), suggesting that both EGC-1 and ELLI-1 localize to uncharted subcompartments of the *C. elegans* germ granule.

Interestingly, the EGO-module factors (EGO-1, DRH-3 and EKL-1) exhibited intriguing subcellular distributions within different subcompartments during germ cell differentiation. For example, like EGC-1 and ELLI-1, the EGO module factors failed to colocalize with the markers of the P, Z, or S compartments of the germ granule at any stage of germ cell development within the adult germline (Supplementary Figs. 3a–d, 7b–d, 8a–c). The EGO module factors constitutively colocalized with markers of E granules (tagRFP::ELLI-1)

throughout the germline and partially colocalized with Mutator foci (mCherry::MUT-16) in the mitotic, transition and late pachytene regions of the germline (Supplementary Fig. 9a–c). In the early/mid pachytene region of the adult germline, the EGO module factors colocalized with components of E granules but rarely with Mutator foci (Supplementary Fig. 9a–c). At the diakinesis stage of germline development, EGC-1 and ELLI-1 did not form observable foci (Supplementary Fig. 9d); however, the EGO module formed foci, and these foci colocalized with markers of Mutator foci (Supplementary Fig. 9e). These data suggest that the EGO module could localize to both the E and M compartments of the germ granule and the relative distribution of the EGO module to these compartments varies across the stages of germ cell development. We summarize the subcellular localization and colocalization patterns of EGO-1 and its interacting partners within the subcompartments of the germ granule (Fig. 4c), and conclude that EGC-1 and ELLI-1 localize to an uncharted subcompartment of the germ granule, which we name the E granule, and the EGO module is distributed between E granules and Mutator foci.

The ordering of the P, Z, and M compartments of the germ granule is not random. For example, the Z granule typically lies directly adjacent to and between one P granule and one Mutator focus[22]. We wondered how the E granule might be positioned relative to the other compartments of the germ granule. To address this question, we generated animals expressing PGL-1::BFP, mCherry::ZNFX-1 and ELLI-1::GFP to simultaneously visualize the P, Z, and E granules. We imaged 210 perinuclear foci and quantified the number of times that a P, Z, or E granule was in contact with the other compartments (*i.e.*, no empty space observable between two foci). In 32% of cases, we observed a single E granule contacting a single P granule, but not a Z granule (Fig. 4d). In these cases, the P and Z granules were in contact, but the E and Z granules were not. In 48% of cases, we observed an E granule contacting both a P granule and a Z granule (Fig. 4d). In 20% of cases, two or more E granules were observed contacting a single P granule, and at least one of the two E granules also contacted a Z granule (Fig. 4d). In summary, the E granule is nearly always found adjacent to a P granule and is sometimes in contact with both a P and Z granule. The E granule is rarely found in contact with a Z granule without also being in contact with a P granule. We conclude that the positioning of the E granule within the larger germ granule is nonrandom (see discussion).

### EGC-1 and ELLI-1 are needed for the accumulation of the EGO module in the E granule

The assembly of many biomolecular condensates is driven by intrinsically disordered/low complexity proteins[67,68]. Among the E granule component proteins, EGC-1 and ELLI-1 possess low-complexity domains (Supplementary Fig. 10)[62], hinting that these proteins might help mediate E granule assembly.

We first examined the subcellular localization of E granule components upon the depletion of EGC-1 or ELLI-1. In *egc-1* mutant, the number of ELLI-1::GFP-labeled foci was strongly reduced to 23% of that observed in wild-type animals (Fig. 5a); in *elli-1* mutant, EGC-1::GFP diffused throughout the cytosol and the nucleus, suggesting that the localization of EGC-1 and ELLI-1 to E granules was interdependent (Fig. 5a). The protein level of ELLI-1 was dramatically reduced in *egc-1(-)* mutants, suggesting that the loss of EGC-1 causes a reduction in ELLI-1 proteins, which might in turn affect the formation of ELLI-1 foci (Fig. 5b). The protein level of EGC-1 was not obviously affected by the loss of ELLI-1, as shown by western blotting, indicating that the loss of EGC-1 foci in animals lacking ELLI-1 was not simply due to the loss of EGC-1 protein (Fig. 5c). EGC-1::GFP and ELLI-1::GFP remained localized to perinuclear foci in animals harboring the hypomorphic *drh-3(ne4253)* allele, in which the function of the EGO module was severely blocked (Fig. 5a)[40]. The localization of EGO module factors (EGO-1, DRH-3 and EKL-1) to germline foci decreased in both *egc-1(-)* and *elli-1(-)* animals; however, residual EGO module-marked foci were still noticeable (Fig. 5d and Supplementary Fig. 11a). The protein levels of EGO-1, DRH-3 and EKL-1 were not obviously affected by the loss of EGC-1 or ELLI-1 (Fig. 5e–g).

Interestingly, upon the loss of EGC-1, the residual EGO module (EGO-1, DRH-3 and EKL-1) foci did not colocalize with the ELLI-1 foci (Fig. 5h). We wondered whether the residual EGO module-marked foci in *egc-1* animals might represent the EGO module localized to other germline granule subcompartments, for example, Mutator foci. The following data support this model. First, the residual EGO module-marked foci in *egc-1(-)* or *elli-1(-)* animals colocalized with tagRFP::MUT-16, which marks Mutator foci (Supplementary Fig. 11b). Second, MUT-16 is needed for assembly of Mutator foci[20]. The residual EGO module-marked foci in *egc-1(-)* or *elli-1(-)* animals were absent in *egc-1;mut-16* and *elli-1;mut-16* double mutant animals (Fig. 5d and Supplementary Fig. 11a). Taken together, the data suggest that EGC-1 and ELLI-1 promote E granule assembly and are needed to recruit the EGO module into E granules.

### The E granule forms independently of other germ granule compartments

The effects of *egc-1* and *elli-1* mutations on the assembly of germ granule compartments appeared specific to the E granule. For example, *egc-1* or *elli-1* mutation did not obviously affect the formation of Z granules (indicated by GFP::ZNFX-1 fluorescence patterns), SIMR foci (indicated by SIMR-1::tagRFP fluorescence patterns), or Mutator foci (indicated by mCherry::MUT-16 fluorescence patterns) (Fig. 5i). *egc-1* or *elli-1* mutations also did not disrupt P granule formation (Fig. 5i). However, for unknown reasons, these mutations did cause an increase in the size of some P granules, which is consistent with a previous report (Fig. 5i and Supplementary Fig. 12a)[62]. Finally, the mutations known to disturb the perinuclear localization of P, Z or M compartments of the germ granule did not affect EGC-1 or ELLI-1's ability to form perinuclear foci (Supplementary Fig. 12b, c). Taken together, the data suggest that the E granule forms independently of the other germ granule subcompartments.

### The E granule promotes the production of a subset of siRNAs in the germline

The RNA-dependent RNA polymerases EGO-1 and RRF-1 synthesize 22G RNAs, which are the most abundant class of endogenous siRNA produced in *C. elegans* and target thousands of germline genes for regulation[29]. To assess whether E granules might contribute to the production of 22G RNAs, we sequenced total small RNAs from wild-type, *egc-1(-)* and *elli-1(-)* animals in a 5′ phosphate-independent method. 22G RNAs were mapped to the *C. elegans* genome, and the number of siRNAs complementary to each *C. elegans* gene was quantified (Fig. 6a, b). Germline-expressed genes targeted by twofold fewer siRNAs in *egc-1(-)* or *elli-1(-)* animals than in wild-type animals were identified. The analysis identified 1504 and 1282 genes whose siRNAs were depleted twofold or more in *egc-1(-)* and *elli-1(-)* animals, respectively (Fig. 6c). Genes whose siRNAs altered in *egc-1(-)* or *elli-1(-)* overlapped extensively (Fig. 6c). For instance, 93% (1192/1282) of genes whose siRNAs decreased in *elli-1* mutants were also depleted of siRNAs in *egc-1* mutants. (Fig. 6c). Henceforth, we refer to siRNAs depleted in *egc-1(-)* or *elli-1(-)* animals as the E-class siRNAs and the genes targeted by these siRNAs as the E-class genes.

We next asked whether the E-class siRNAs were, as expected, dependent on EGO-1. We sequenced siRNAs from *ego-1(om84)* mutant animals, and siRNAs depleted > 2-fold in these mutant animals were identified (Supplementary Fig. 13a, b). Indeed, 1521/1594 of the E-class siRNAs were also depleted in *ego-1* mutant animals (Fig. 6d). We conclude that EGC-1 and ELLI-1 regulate the biogenesis and/or stability of a similar suite of EGO-1-dependent 22G RNAs.

### Different germ granule compartments produce different siRNA populations

To compare the siRNAs generated in different subcompartments of the germ granules, we also sequenced siRNAs from *mut-16(-)* animals, which lack Mutator foci. We identified 1699 genes whose siRNAs depleted > 2-fold in *mut-16(-)* animals (henceforth, M-class siRNAs). Our list of M-class siRNAs overlapped extensively (approximately 75%) with lists of M-class siRNAs generated in previous studies, suggesting the reliability of our dataset (Supplementary Fig. 13c)[24,69]. We compared our E-class and M-class siRNA lists and found that E-class and M-class siRNAs targeted distinct gene sets (Fig. 6e). Metagene analysis confirmed that E-class siRNAs were generally unaffected in *mut-16(-)* animals that lacked Mutator foci (Fig. 6f) and that the M-class siRNAs were unaffected in animals that lacked E granules (Fig. 6g and Supplementary Fig. 13d). For example, siRNAs targeting *bath-45*, *c38d9.2* and *f15d4.5* were MUT-16-dependent but were unaffected in *egc-1(-)* or *elli-1(-)* animals (Fig. 6h). *cls-2*, *F01G4.4* or *tebp-2* siRNAs depended on EGC-1 and ELLI-1 but were unaffected in *mut-16(-)* animals (Fig. 6h). The data suggest that E granules are responsible for synthesizing siRNAs that are largely distinct from those synthesized in Mutator foci.

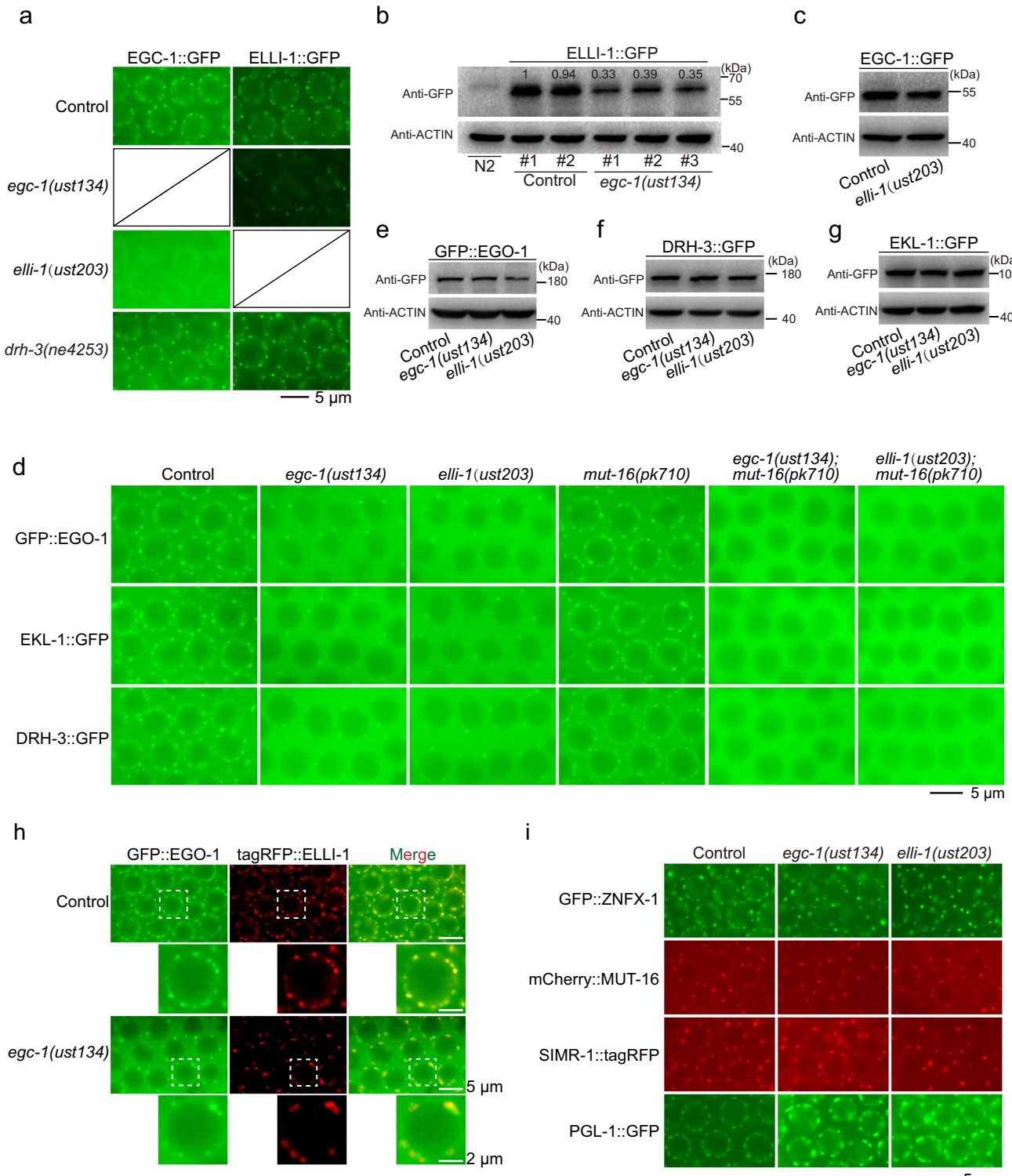

**Fig. 5 | EGC-1 and ELLI-1 are required for the accumulation of the EGO module in the E granule. a** Pachytene germ cells of animals that express EGC-1::GFP or ELLI-1::GFP in the indicated animals. The *drh-3(ne4253)* lesion (T834M) alters a residue that contacts RNA in the Vasa crystal structure[40]. The hypomorphic *drh-3(ne4253)* allele was reported to cause RNAi defective phenotype, siRNA biogenesis defects and dramatically reduced brood size at 20 °C[40]. The indicated animals were grown at 20 °C. **b**, **c** Western blotting of the indicated GFP-tagged proteins in the indicated animals. Actin was used as a loading control. Each experiment has been repeated three times with similar results. Source data are provided as a Source Data file. **d** Pachytene germ cells of animals that express GFP::EGO-1, EKL-1::GFP or DRH-3::GFP in the indicated animals. The perinuclear localization of EGO-1, EKL-1 and DRH-3 are dramatically decreased upon the loss of EGC-1 or ELLI-1 and are completely blocked

in *egc-1;mut-16* or *elli-1;mut-16* double mutant animals. **e**, **f**, **g** Western blotting of the indicated GFP-tagged proteins in the indicated animals. Actin was used as a loading control. Each experiment has been repeated three times with similar results. Source data are provided as a Source Data file. Defects in E granule assembly did not affect the expression levels of the EGO module components (EGO-1, DRH-3 and EKL-1). **h** Fluorescence micrographs of the indicated animals expressing GFP::EGO-1 and tagRFP::ELLI-1. **i** Fluorescence micrographs of pachytene germ cells from animals expressing GFP::ZNFX-1, mCherry::MUT-16, SIMR-1::tagRFP and PGL-1::GFP in the indicated animals. All images were taken by a Leica upright DM4 B microscope equipped with a Leica DFC7000 T camera and an HC PL APO 100x/1.40-0.70 oil objective. All images are representative of more than three animals.

*C. elegans* 22G RNAs can be grouped into two major classes (CSR-1-class and WAGO-class) based upon the Argonaute proteins to which they bind[29]. Current models posit that CSR-1-class siRNAs are synthesized by the RdRP EGO-1 and that WAGO-class siRNAs are likely synthesized by the Mutator foci-localized RdRP RRF-1[29]. Because EGO-1 localizes to E granules, we tested whether the E-class siRNAs were likely to be CSR-1-class siRNAs. Indeed, a comparison of our list of E-class siRNAs with the published lists of CSR-1 bound siRNAs showed that the E-class siRNAs represented a subset of the CSR-1-class siRNAs (Supplementary Fig. 14a). Consistent with previous reports, the M-class siRNAs are largely WAGO-class siRNAs (Supplementary Fig. 14b)[20,69]. The following examples illustrate these points: E-class siRNAs targeting the E-class genes *cls-2*, *F01G4.4* and *tebp-2* are enriched in CSR-1 IP samples[41], and M-class siRNAs targeting the M-class genes *bath-45*, *c38d9.2* and *f15d4.5* are enriched in WAGO-1 IP samples (Supplementary Fig. 14c)[40]. Taken together, the data suggest that siRNAs produced in the E and M compartments of the germ granule bind distinct AGOs and that different germ granule subcompartments specialize in producing distinct, largely nonoverlapping populations of small regulatory RNAs.

## EGC-1 and ELLI-1 coordinate the production of a subset of Mutator foci-derived siRNAs

siRNA-seq analysis identified 199 genes for which the mapped siR-NAs increased in abundance in *egc-1* or *elli-1* mutants (Fig. 7a and Supplementary Fig. 15a). The upregulated siRNAs in the *egc-1* and *elli-1* mutants exhibited a pronounced overlap (Supplementary Fig. 15b). For example, siRNAs targeting *T03D3.5, T16G12.4, sid-1* and *rde-11* were increased more than 10 times in both *egc-1* and *elli-1* mutants (Fig. 7b). Since defects in E granule assembly did not block the localization of the EGO module in Mutator foci (Supplementary Fig. 11a) and the EGO module could accumulate in Mutator foci in the early and middle pachytene regions of the germline in *egc-1* or *elli-1* mutants (Fig. 5c and Supplementary Fig. 11b), we wondered whether these upregulated siRNAs were produced in Mutator foci or other places. Indeed, the upregulated siRNAs in *egc-1* or *elli-1* mutants largely belong to the M-class siRNAs, suggesting that the depletion of EGC-1 or ELLI-1 may enhance the production of a subset of Mutator foci-derived siRNAs (Fig. 7c). We further sequenced siRNAs from *egc-1 or elli-1* animals that also harbored a mutation in *mut-16* and, therefore, lacked both E granules and Mutator foci. siRNAs that increased in abundance in *egc-1* or *elli-1* mutants were abolished in *egc-1;mut-16* and *elli-1;mut-16* double mutant animals (Fig. 7d), suggesting that this group of siRNAs was likely produced in Mutator foci.

EGC-1 and ELLI-1 are required for feeding RNAi response. We deep-sequenced siRNAs upon RNAi targeting the *gfp* gene in animals expressing germline GFP::H2B. We found that the production of *gfp* siRNAs was dramatically prohibited in *egc-1* and *elli-1* mutants (Supplementary Fig. 15c). Since the Mutator pathway does not seem to be affected in *egc-1* or *elli-1* mutants, one hypothesis is that their defects in exogenous RNAi response are indirectly caused by misexpression of RNAi related genes. Recent studies have reported that *meg-3/4* animals produce aberrant siRNAs targeting *sid-1* and *rde-11*, which silence the expression of the two genes and consequently result in defects in feeding RNAi response[50–52]. We found that siRNAs targeting the two RNAi-related genes, *sid-1* and *rde-11*, were both dramatically upregulated in *egc-1* and *elli-1* mutants (Fig. 7b). Furthermore, several RNAi-related genes, including the *rde-11* gene, were misexpressed upon a hypomorphic mutation in the *elli-1* gene[62]. Thus, we performed mRNAs deep-sequencing of wild-type, *egc-1*, *elli-1* and *mut-16* animals. As expected, *sid-1* and *rde-11* mRNA levels were dramatically down-regulated in *egc-1* and *elli-1* mutants (Fig. 7e, f and Supplementary Fig. 15d, e). qRT–PCR analysis further confirmed the downregulation of *sid-1* and *rde-11* mRNAs in the mutants (Supplementary Fig. 15f). As SID-

1 and RDE-11 are involved in dsRNA transportation and siRNA biogenesis during the RNAi response[70–72], these data suggested that the silencing of the *sid-1* and *rde-11* genes may underlie the defects of *egc-1* and *elli-1* animals in exogenous RNAi.

Together, these data suggested that the assembly of E granules may coordinate the production of a subset of Mutator foci-derived siRNAs to promote feeding RNAi response.

## Most E-class genes are not desilenced in *egc-1* or *elli-1* mutants

The E-class siRNAs were mainly bound to CSR-1, which is thought to possess multiple gene regulatory functions in the germline[73–76]. To examine the effects of E granule-dependent siRNAs on the target mRNAs, we analyzed mRNA-seq data in wild-type, *egc-1(-)*, *elli-1(-)* and *mut-16(-)* animals. A cutoff criterion of a 2.0-fold change was applied for filtering of differentially expressed mRNAs. We did not observe a substantial overall change in the mRNA expression levels of E-class genes in *egc-1(-)* and *elli-1(-)* animals, compared with those in wild-type animals (Supplementary Fig. 16a). For example, the accumulation of *klp-7* and *cls-2* mRNAs was unchanged in both *egc-1* and *elli-1* mutants (Supplementary Fig. 16b). Among the 1594 E-class genes, only 12 genes were upregulated in both *egc-1(-)* and *elli-1(-)* animals, but not in *mut-16(-)* animals (Supplementary Fig. 16c), implying that E granule-derived siRNAs might not markedly regulate the expression of endogenous genes.

EGO-1 is needed for the generation of E-class siRNAs and has been reported to modulate gene expression via the production of mRNA-templated siRNAs[54]. We performed mRNA deep-sequencing of the *ego-1* mutant (Supplementary Fig. 16d), and found that 165 E-class genes were significantly upregulated in the *ego-1* mutant (Supplementary Fig. 16d). EGO-1-targeted genes that have been previously reported were also identified in our dataset[54], for example, *F01G4.4* (3.7 fold), *klp-7* (6.4 fold), *cec-6* (5.5 fold) and *tebp-2* (8.5 fold). Interestingly, these 165 E-class genes were roughly unaffected in *egc-1* and *elli-1* mutants (Supplementary Fig. 16e). We tested a number of E-class genes by qRT–PCR and found that the expression levels of E-class genes depend on EGO-1, but not EGC-1 and ELLI-1, hinting their different degrees of necessity in promoting the generation of E-class siRNAs, although E-class siRNAs were all significantly reduced among *egc-1*, *elli-1* and *ego-1* mutants (Supplementary Fig. 16f).

## The E granule promotes the synthesis of 5′ siRNAs

Visual inspection of E-class siRNA mapping to E-class genes revealed that the E-class siRNAs lost in *egc-1* or *elli-1* mutants were predominantly mapped to the 5′ portion of the E-class genes. For example, siRNAs targeting *cls-2*, *klp-7*, *F01G4.4*, *tebp-2*, *csr-1*, *ama-1*, and *hcp-1* were evenly distributed across the length of their target mRNAs in wild-type animals (Fig. 8a and Supplementary Fig. 17a). In *egc-1* and *elli-1* mutants, however, siRNAs mapping to the 5′, but not 3′-most, portions of these E-class genes were depleted (Fig. 8a and Supplementary Fig. 17a). The siRNA targeting the 3′ most of the E-class genes relied on EGO-1, but not EGC-1 and ELLI-1, and typically include the last exon of the E-class genes (Fig. 8a and Supplementary Fig. 17a). Metagene analysis showed that the loss of siRNAs mapping to the 5′ portion, but not the 3′ end siRNAs, of E-class genes was a general consequence in *egc-1* and *elli-1* mutants (Fig. 8b). Therefore, the two groups of siRNAs were termed the E-class 5′ siRNAs and the E-class 3′ siRNAs respectively. The data show that EGC-1 and ELLI-1 are needed for synthesizing and/or stabilizing siRNAs derived from the 5′ portions of the E-class mRNAs and that EGO-1 is needed for both the 5′ portions and the 3′ siRNAs.

Since the mutation of EGC-1 and ELLI-1 depleted E granules but not the production of E-class 3′ siRNAs, we asked whether E-class 3′ siRNAs are generated in other cellular subcompartments, for example, Mutator foci, or in the cytosol. We then analyzed siRNAs in

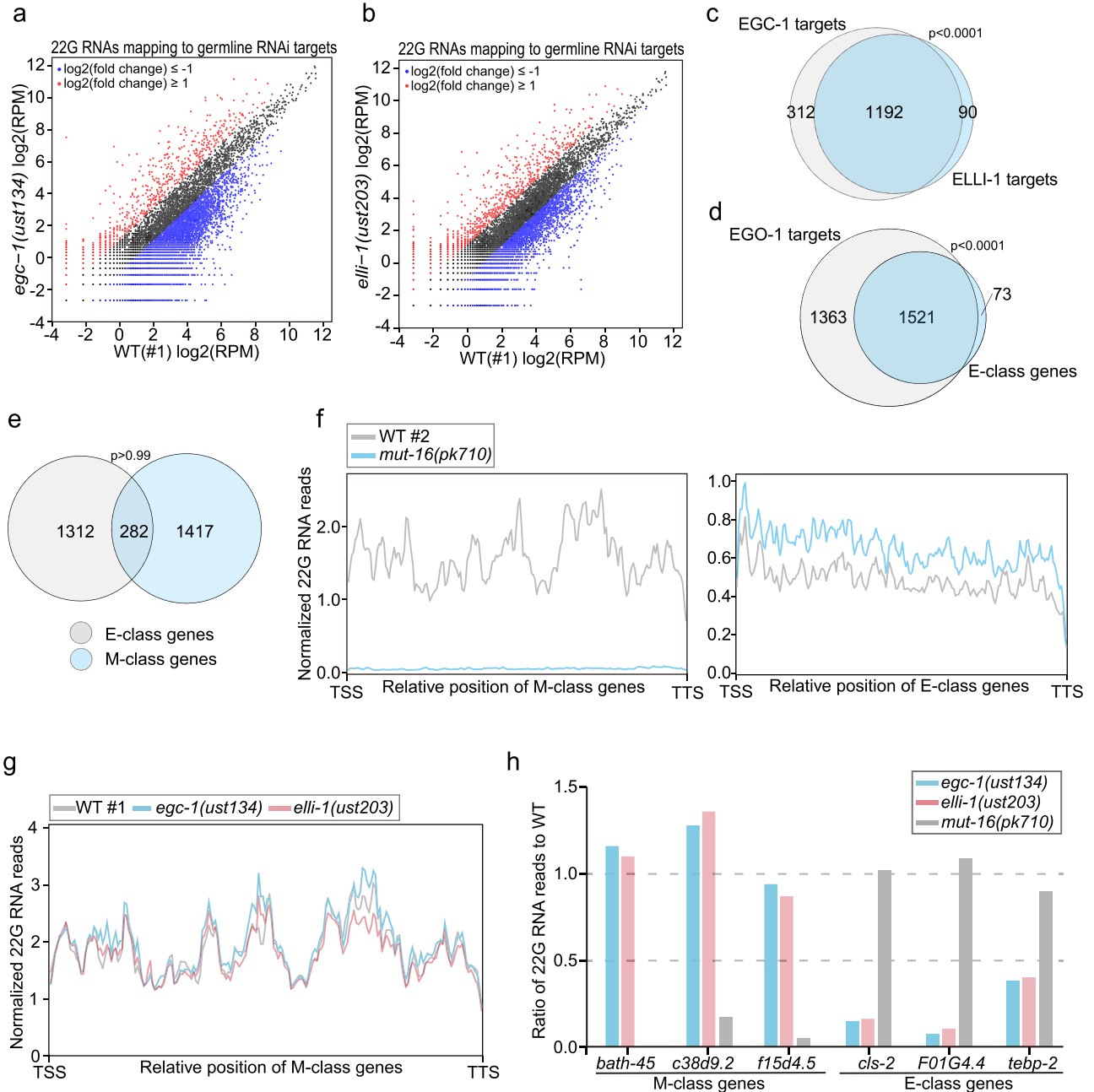

**Fig. 6 | The E granule promotes the production of a subset of siRNAs in the germline. a, b** Scatter plots showing gene-by-gene comparisons of normalized siRNA abundances. siRNAs from wild-type, *egc-1(-)* and *elli-1(-)* animals were sequenced in a 5′ phosphate-independent method. 22G RNAs were mapped to the *C. elegans* genome and the number of reads complementary to each *C. elegans* gene was quantified. A cutoff criterion of a 2-fold change was applied to identify differentially expressed siRNAs. Genes with upregulated and downregulated siRNAs are shown in red and blue, respectively. **c** Proportional Venn diagram showing comparisons between sets of genes that are the targets of 22G RNAs in the indicated animals. Genes yielding ≥ 10 siRNA reads per million total small RNA reads (RPM) in wild-type animals, were selected for analysis. Soma-enriched siRNA targets were excluded from the above analysis[40]. Overlaps between *egc-1(ust134)* and *egc-1(ust206)* are annotated as EGC-1 targets; overlaps between *elli-1(ust203)* and *elli-1(ust204)* are annotated as ELLI-1 targets. Genes targeted by twofold fewer siRNAs in *egc-1(-)* or *elli-1(-)* animals than in wild-type animals were identified as the E-class genes. **d** Proportional Venn diagram showing the overlap among EGO-1 siRNA targets and E-class genes. **e** Proportional Venn diagram showing the overlap among E-class genes and M-class genes. A list of E- and M-class genes is shown in Supplementary data 2. **f** Metaprofile analysis showing the distribution of normalized 22G RNA (sRNA-seq) reads (RPM) along M-class genes and E-class genes in control animals and *mut-16* mutants. The metaprofiles were generated according to the method described previously in ref. 77. **g** Metaprofile analysis showing the distribution of normalized 22G RNA (sRNA-seq) reads (RPM) along M-class genes in the indicated animals. The loss of EGC-1 or ELLI-1 does not overall affect the production of 22G RNAs mapping to the M-class genes. **h** Ratio of normalized 22G RNA reads from a representative subset of E-class genes and M-class genes in the indicated mutants to wild-type animals (WT = 1.0). *bath-45*, *c38d9.2* and *f15d4.5* are M-class genes; *cls-2*, *F01G4.4* and *tebp-2* are E-class genes.

*egc-1;mut-16* and *elli-1;mut-16* double mutant animals. As expected, M-class 22G RNAs were completely depleted in the double mutants (Supplementary Fig. 17b, c). However, the 22G RNAs mapping to 3′ end regions were unaffected in *egc-1;mut-16* and *elli-1;mut-16* double mutants (Fig. 8c, d and Supplementary Fig. 17d). Because both E granules and Mutator foci were disrupted in the double mutants, the presence of the E-class 3′ siRNAs suggested that they might be generated by the cytosol-localized EGO module (see discussion).

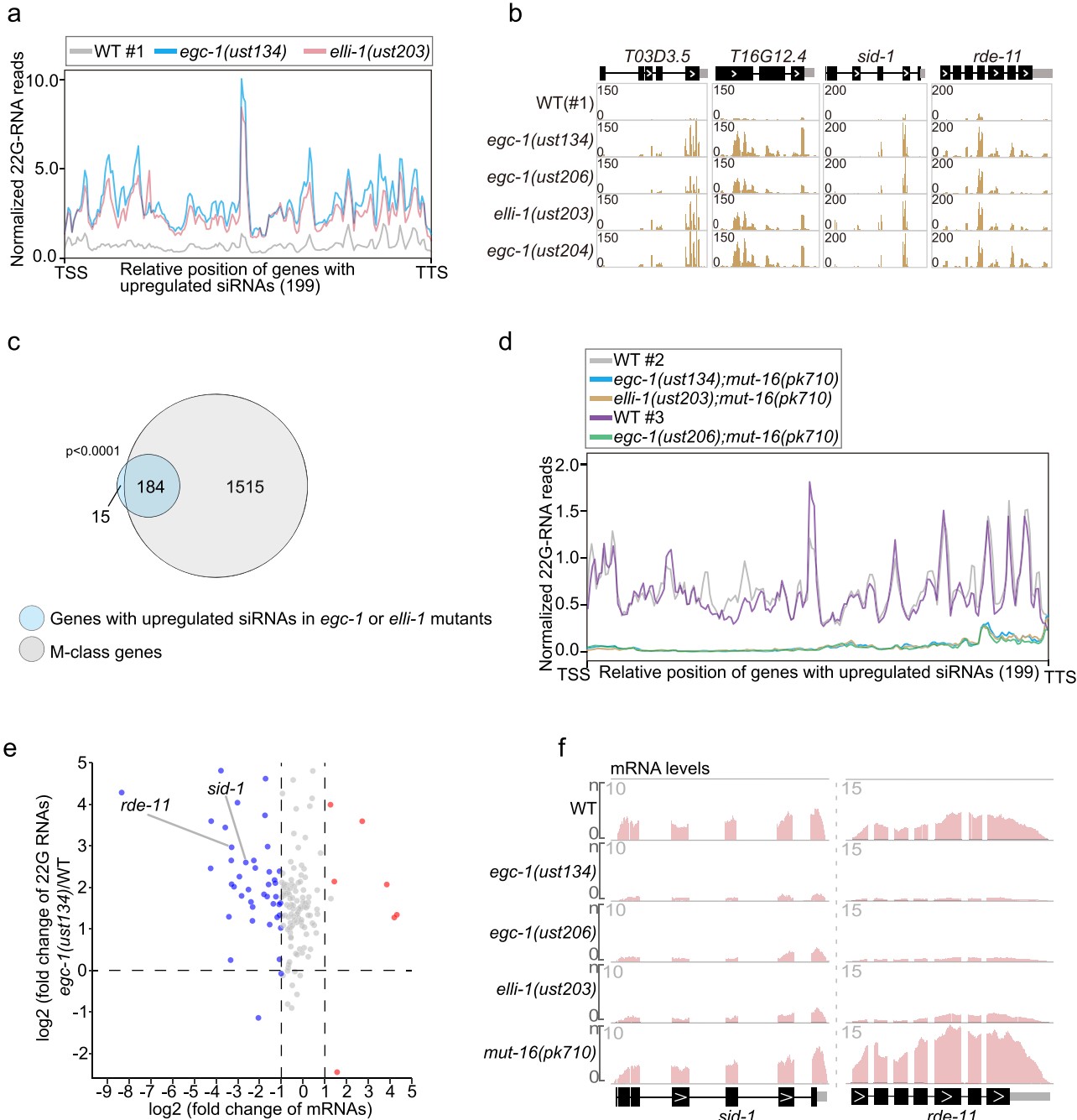

**Fig. 7 | EGC-1 and ELLI-1 coordinate the production of a subset of Mutator foci-derived siRNAs to regulate gene expression. a** Metaprofile analysis showing the distribution of normalized 22G RNA (sRNA-seq) reads (RPM) along 199 genes with upregulated siRNAs in *egc-1* or *elli-1* animals. **b** Normalized 22G RNA read distribution across *T03D3.5, T16G12.4, sid-1* and *rde-11* in the indicated animals. **c** Proportional Venn diagram showing the overlap among M-class genes and 199 genes with upregulated siRNAs in *egc-1* or *elli-1* mutants. **d** Metaprofile analysis showing the distribution of normalized 22G RNA (sRNA-seq) reads (RPM) along 199 genes with upregulated siRNAs in *egc-1* or *elli-1* mutants in the indicated animals. **e** Volcano plot showing the fold-change of mRNAs (x-axis) versus the fold-change of siRNAs (Y-axis) of the above 199 genes. *sid-1* and *rde-11* are indicated. **f** Normalized mRNA read distribution across *sid-1* and *rde-11* in the indicated animals.

Interestingly, we reanalyzed published datasets and found that the depletion of CSR-1, or mutations in CSR-1 that abolish its slicer activity, also led to a loss of the 5′ E-class siRNAs without affecting the 3′ E-class siRNA production, hinting at a stepwise synthesis of E-class siRNAs along with sequential processing of E-class mRNA templates (Supplementary Fig. 17e–h)[41,77]. Taken together, the data suggest that the E granule promotes the production of a subset of germ cell siRNAs, which are synthesized from the 5′ portion of a subset of germline mRNAs, while siRNAs from the 3′ end of these mRNA templates may be produced by the cytosol-localized EGO module (see discussion).

## Discussion

Here we identified five proteins (EGC-1, ELLI-1, EGO-1, DRH-3, and EKL-1) that localize to a distinct subcompartment of the *C. elegans* germ granule, which we named the E granule. We found that the intrinsically disordered proteins EGC-1 and ELLI-1 promote E granule assembly and that the E granule is nonrandomly positioned within

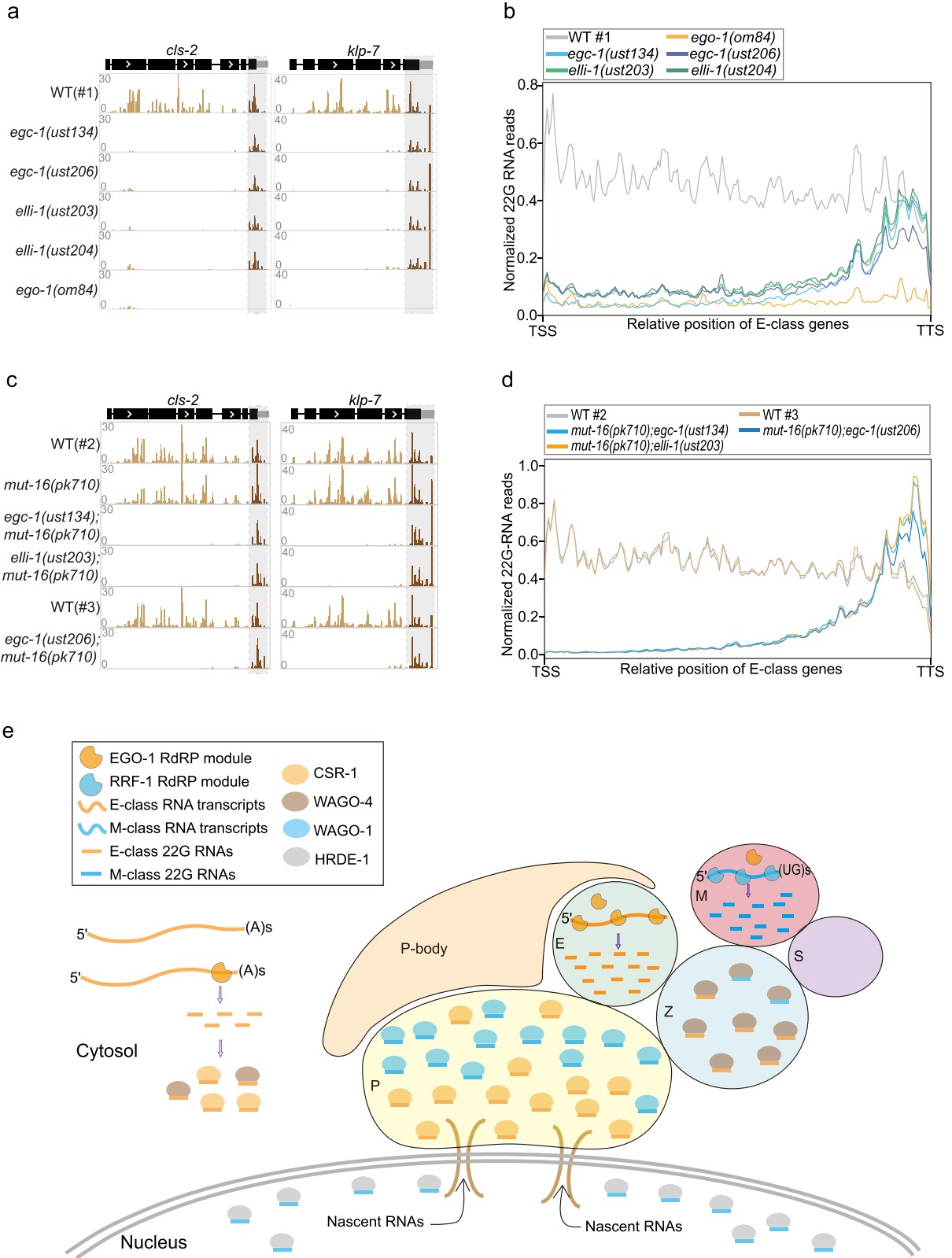

the larger granule, with respect to the other subcompartments. Our data suggest that E granules and Mutator foci produce distinct sets of small RNAs and that the E granule specializes in the production of siRNAs derived from the 5′ termini of CSR-1-class mRNAs. Thus, our results define a distinct subcompartment of the *C. elegans* germ granule and identify five proteins localizing to this compartment. Our results suggest that one biological function of germ granule

compartmentalization is to coordinate localized production of specialized classes of small regulatory RNAs.

## Components of the E granule

We identify five proteins that localize to the E granule. Two of these proteins (EGC-1 and ELLI-1) possess low-complexity domains and are needed for E granule assembly[62]. Given that many low-complexity

**Fig. 8 | The E granule promotes specialized synthesis of 5′ siRNAs. a** Normalized 22G RNA read distribution across E-class siRNAs targeting *cls-2* and *klp-7*. Additional examples of 22 G RNAs covering E-class genes can be found in Supplementary Fig. 17a. **b** Metaprofile analysis showing the distribution of normalized 22G RNA (sRNA-seq) reads (RPM) along E-class genes in the indicated animals. EGC-1 and ELLI-1 are exclusively required for the production of 5′ E-class siRNAs. **c** Normalized 22G RNA read distribution across E-class siRNA-targeted genes *cls-2* and *klp-7*. **d** Metaprofile analysis showing the distribution of normalized 22G RNA (sRNA-seq) reads (RPM) along E-class genes in the indicated animals. **e** A working model for the role of germ granule compartmentation in siRNA generation and AGO/siRNA function. E granules and Mutator foci are two independent subcompartments of perinuclear germ granules for 22G RNAs generation using a largely nonoverlapping set of RNA transcripts as templates. E granules-derived 22G RNAs are bound to CSR-1, while 22G RNAs derived from Mutator foci are bound to WAGO-1 and HRDE-1. WAGO-4 binds to siRNAs derived from the exogenous RNAi treatment and promotes transgenerational inheritance[22,91]. It is currently unknown whether these siRNAs are produced by the Mutator complex in Mutator foci or by EGO-1 in E granules. 22G RNAs mapping to the 3′ regions of E-class mRNAs are likely produced by the EGO module in the cytosol. Different AGO/siRNAs complexes localize to distinct intracellular subcompartments of the germ granule. Germline P-bodies were reported to be located on top of P granules[84]. The relative sizes of different subcompartments of germ granules are not yet known.

domain proteins have been linked to biomolecular condensate formation[78–80], we speculate that one function of EGC-1 and ELLI-1 is to help assemble the E granule and recruit the EGO module into the E granule. There are more than 90 *C. elegans* proteins known to be germ granule enriched[16]. A recent study using the Turbo ID technology identified the proteome in P granules[56]. Using the TurboID system to identify E granule components may further help to decipher E granule biological functions.

## The compartmentalization of the germ granule

Cells organize many of their biochemical reactions within non-membrane organelles termed biomolecular condensates[2]. The *C. elegans* germ granule is subdivided into distinct subcompartments, which house distinct proteins and, likely, distinct RNA constituents[16]. How and why the germ granule is compartmentalized is largely a mystery. Here, we identify a distinct compartment of the germ granule that we term the E granule and show that animals that fail to assemble E granules exhibit defects in germline RNAi and fertility (Fig. 2b–f). The data suggest that the subdivision of the germ granule into distinct regions, such as the E granule, is important for germ cell function. Interestingly, we find that E granule assembly is; 1) developmentally regulated (Supplementary Figs. 2a–d, 5c–h); and 2) nonrandomly positioned with respect to the other subcompartments of the germ granule (Fig. 4d). Thus, E granule assembly is spatiotemporally regulated. Because we find that the E granule is needed for producing a subset (E-class) of germline siRNAs (Fig. 6a–c), we speculate that the spatiotemporal regulation of E granule assembly enables germ cells to generate E-class siRNAs in specific regions and at specific stages of development to promote germ cell function. We propose a model outlining a possible architecture of the *C. elegans* germ granule subcompartments and a potential function for these subcompartments in siRNA synthesis and germ cell function (Fig. 8e). Technical advances that enable simultaneous imaging of these compartments of the germ granule will be essential to assess the global architecture of the germ granule subcompartments.

The nonrandom positioning of the E granule within the germ granule resembles previous reports showing that the P, Z, M, and S compartments of the germ granule are nonrandomly positioned[22,24]. The P and Z compartments of the germ granule are known to exhibit liquid-like properties[9,22]. Although we have not yet tested this idea, it seems reasonable to speculate that the E, M, and S compartments of the germ granule may also behave like liquids. If true, this raises the fascinating question of how and why multiple condensates could be arranged in an ordered manner across space and time, and perform complex mixing and demixing. In germ cells, germ granules localize adjacent to nuclear pores where they are thought to act as sites of mRNA surveillance[11]. The protein constituents of the P and Z granules demix to form distinct foci concomitantly with the advent of germline transcription during development, hinting that the passage of mRNAs through germ granules may contribute to the demixing of the P and Z granules[11,22,81]. In fact, RNA transcripts have been reported to promote the formation of germ granule subcompartments. For example, the injection of young adult gonads with the transcriptional inhibitor a-amanitin induced pachytene-specific loss of PGL-1 foci and MUT-16 foci[81]. RNAs transcribed in the nucleus pass through several specific cellular structures, including the nuclear pore complex and the germ granule, and are finally transported to the cytoplasm or the common cytoplasmic core of the germline for translation in *C. elegans*[11,82]. Defects in these transportation processes usually elicit disordered perinuclear germ granule architecture. For example, the loss of a series of nuclear pore complex proteins and the depletion of mRNA exporting factors, such as IMB-4, IMB-5, RAN-1 and RAN-4, elicits the diffusion of P granules into the cytosol[83]. Interestingly, a recent study reported that the depletion of two conserved P-body components, which bind to mRNAs and are required for the regulation of translation, elicited the fusion of the P and Z granules in germ cells[84]. Thus, we speculate that proper RNA flows in cells, including transportation and translation, may promote the perinuclear localization or the formation of multiphase architecture of germ granules. Systemic investigation of the influence of abnormal RNA transport and metabolism on germ granule architectures may help to decipher how RNA transcripts contribute to the establishment of these multicompartmentalized germ granules.

## E-class siRNA

Mutations in *egc-1* and *elli-1* result in a failure of animals to recruit the EGO module to E granules (Fig. 5c, d and Supplementary Fig. 11a, b), and this failure to recruit the EGO module is associated with a loss of the E-class siRNA, which is antisense to the 5′ regions of CSR-1-class mRNAs (Fig. 8b). Thus, the data suggest that the EGO module synthesizes the E-class 5′ siRNA in the E granule. siRNAs mapping to the 3′ portions of CSR-1 mRNAs are, like the 5′ siRNAs, dependent on EGO-1 (Fig. 8b). However, the 3′ siRNAs are not dependent on the presence of E granules or Mutator foci (Fig. 8d), suggesting that the siRNAs mapping to the 3′ regions of CSR-1-class mRNAs may be produced by EGO-1 in the cytosol.

Interestingly, the deletion of CSR-1, or the mutation in CSR-1 that inhibit CSR-1's slicer activity, also leads to a loss of the 5′ CSR-1 siRNAs without affecting the production of 3′ CSR-1 siRNAs[77]. Taken together, the data hint that the EGO module may produce 3′ siRNAs in the cytosol using CSR-1-class mRNA transcripts as templates. These 3′ siRNAs then bind CSR-1 and target additional CSR-1-class mRNAs, which are transported into E granules for amplification by the EGO module. Alternatively, both 5′ siRNAs and 3′ siRNAs may be generated in E granules in wild-type animals and cells may take unknown measures to sustain 3′ siRNAs production in the cytosol upon defective E granule assembly. The fact that the 3′ siRNAs are made outside E granules in *egc-1* or *elli-1* mutants does not necessarily mean they are equally made outside in wild-type animals. Further work is needed to understand how and why the EGO module might produce both 5′ and 3′ CSR-1-class siRNAs and whether and how they might do so in distinct regions of the cell.

siRNAs regulate the expression levels of targeted genes in a sequence-specific manner in a variety of organisms. However, most E-class genes are not desilenced in *egc-1* or *elli-1* mutants (Supplementary Fig. 16a, b, e, f). The E-class siRNAs are a subset of the

previously identified CSR-1 siRNAs. CSR-1-class siRNAs are thought to possess multiple gene regulatory functions in the germline[29]. For instance, CSR-1-class siRNAs are thought to positively regulate (license) the expression of thousands of germline mRNAs, protecting these genes from piRNA-mediated gene silencing[73,74]. Additionally, CSR-1-class siRNAs are reported to negatively regulate maternal mRNAs during embryogenesis[75,76]. How CSR-1 and its associated siRNAs might protect some germline mRNAs from piRNA silencing while promoting the degradation of other mRNAs is not yet known. It is possible that the two classes of CSR-1 siRNAs, 5′ siRNAs synthesized in the E granule and 3′ siRNAs synthesized in the cytosol, underlie this functional dichotomy. Additional studies are needed to assess if the 5′ and 3′ CSR-1 siRNAs explain the pro- and anti-silencing functions of CSR-1 in the germline.

## Methods

### C. elegans strains

The Bristol strain N2 was used as the standard wild-type strain. All strains were grown at 20 °C unless otherwise specified. The strains used in this study are listed in Supplementary data 3. To collect homozygous *ego-1(om84)* mutant worms, balanced worms were synchronized, and approximately 1000 homozygous mutants were collected when they reached the young adult stage.

### Construction of transgenic strains

For in situ expression of GFP::EGO-1, mCherry::EGO-1, DRH-3::GFP::3xFLAG, EKL-1::GFP::3xFLAG, EGC-1::GFP::3xFLAG, ELLI-1::3xFLAG::GFP, RRF-1::tagRFP, mCherry::MUT-16, SIMR-1::tagRFP, mCherry::ZNFX-1, PGL-1::tagBFP and NPP-9::mCherry, the coding regions of *gfp, 3xflag::gfp, mCherry, tagRFP* or *tagBFP* fused to a linker sequence (GGAGGTGGAGGTGGAGCT) were inserted upstream of the stop codon or downstream of the initiation start codon using the CRISPR/Cas9 system. Plasmids containing repair templates were generated using a ClonExpress MultiS One Step Cloning Kit (C113-02, Vazyme). The injection mix contained pDD162 (50 ng/mL), a repair plasmid (50 ng/mL), pSG259 (*myo-2p::gfp::unc-54utr*) (5 ng/mL) and two or three sgRNAs targeting sequences proximal to the N-termini or C-termini of the genes (each sgRNA plasmid, 20 ng/mL). The mix was injected into adult animals. Three to four days later, F1 worms expressing pharyngeal GFP were isolated under a Leica M165 FC fluorescence stereomicroscope. For GFP- and tagBFP-tagged transgenes, F1 adult worms expressing pharyngeal GFP were picked onto microscope slides, and the GFP fluorescence signals from germ cells were observed under a Leica DM4 B microscope. Worms with observable green fluorescence within the germline were transferred from the slides onto individual NGM plates to lay F2 worms. Then, 16 F2 adult worms were singled onto individual NGM plates, and the homozygous transgenes were subsequently identified by evaluating fluorescence signals in F3 animals and genotyping. For tagRFP- or mCherry-tagged transgenes, F1 worms expressing pharyngeal GFP were isolated under a Leica fluorescence stereomicroscope and transferred onto individual NGM plates to lay F2 animals. The targeted animals with tagRFP or mCherry insertions were screened by PCR.

For ectopically expressed transgenes carrying *mex-5p::tagRFP::mut-16::tbb-2_3'UTR, mex-5p::tagRFP::elli-1::tbb-2_3'UTR* or *mex-5p::tagRFP::elli-1::elli-1_3'UTR*, the DNA elements were integrated into *C. elegans* chromosome I (LG I, -5.51 cM) by CRISPR/Cas9 gene editing together with a *rps-11p::hyg::unc-54_3'UTR* element as previously reported[85]. The targeted worms were screened as previously reported[85]. For ectopically expressed transgenes carrying *ego-1p::3xflag::gfp::ego-1::ego-1_3'UTR* and *wago-1p::3xflag::gfp::wago-1::wago-1_3'UTR*, these elements were integrated into *C. elegans* chromosome II (*ttTi5605* locus) by the MosSCI method[86]. For the ectopically expressed transgene carrying *mex-5p::elli-1::gfp::tbb-2_3'UTR*, the *elli-1* element was immediately inserted downstream of the start codon in the ectopic *mex-5p::gfp::tbb-2_3'UTR* transgene via the

CRISPR/Cas9 method. The above plasmids containing repair templates were generated using the ClonExpress MultiS One Step Cloning Kit (C113-02, Vazyme).

### Construction of mutant strains

To construct sgRNA expression vectors, the 20 bp *unc-119* sgRNA guide sequence in the *pU6::unc-119* sgRNA(F + E) vector was replaced with different sgRNA guide sequences. Plasmid mixtures containing 30 ng/μl of each of the three or four sgRNA expression vectors, 50 ng/μl pDD162 plasmid, and 5 ng/μl pSG259 were coinjected into wild-type animals. Animals with gene deletions were screened by PCR as described previously[60]. Each homozygous mutant was outcrossed at least 3 times with N2 worms for the elimination of putative off-target mutations introduced by Cas9.

### Immunoprecipitation followed by mass spectrometry analysis

IP-MS was conducted as previously reported[42]. Mixed-stage transgenic worms expressing ectopic GFP::EGO-1 were collected and resuspended in equal volumes of 2× lysis buffer (50 mM Tris-HCl [pH 8.0], 300 mM NaCl, 10% glycerol, 1% Triton X-100, Roche ®cOmplete EDTA-Free Protease Inhibitor Cocktail, 1 mM PMSF and 10 mM NaF) and lysed in a FastPrep-24 5G homogenizer. The lysate supernatant was incubated with in-house-prepared anti-GFP beads for one hour at 4 °C. The beads were then washed three times with cold lysis buffer. The GFP immunoprecipitates were eluted with chilled elution buffer (100 mM glycine-HCl [pH 2.5]). Approximately 1/8 of each eluate was subjected to western blot analysis. The rest of each eluate was precipitated with TCA or cold acetone and dissolved in 100 mM Tris (pH 8.5) with 8 M urea. Proteins were reduced with TCEP, alkylated with 10 mM IAA, and finally digested with trypsin at 37 °C overnight. LC–MS/MS analysis of the resulting peptides and MS data processing approaches were conducted as previously described in ref. 87. A WD scoring matrix was used to identify high-confidence candidate interacting proteins. The proteins identified in the EGO-1 IP are listed in Supplementary data 1.

### RNAi

RNAi experiments were carried out at 20 °C by placing synchronized embryos onto feeding plates as previously described in ref. 88. *pos-1* and *mex-3* RNAi colonies were obtained from the Ahringer library and sequenced to verify their identity. The *gfp* RNAi clone was obtained from the Fire Laboratory.

### Brood size

L3 worms were placed individually onto fresh NGM plates. The numbers of progeny that reached the L2 or L3 stage were scored. The *ego-1(om84)* is balanced by a *hT2[bli-4(e937) let-?(q782) qIs48]* chromosome and homozygous *ego-1(om84)* worms were singled by selecting worms without pharyngeal GFP under a Leica M165 FC fluorescence stereomicroscope.

### Western blotting

Synchronized young adult worms incubated at 20 °C were collected and washed three times with 1 × M9 buffer. Samples were stored at − 80 °C before use. The worms were suspended in 1 × SDS loading buffer and then heated in a metal bath at 95 °C for 5 ~ 10 min. Then, the suspensions were centrifuged at 17,000 g, and the supernatants were collected. Proteins were separated by SDS–PAGE on gradient gels (10% separation gel, 5% spacer gel) and transferred to nitrocellulose membranes. After washing with TBST buffer (Sangon Biotech, Shanghai) and blocking with 5% milk-TBST, the membranes were incubated with primary antibodies for two hours at room temperature (listed below). After 3 × 10 min washes in TBST, primary antibodies were detected with HRP-conjugated goat anti-rabbit or goat anti-mouse secondary antibodies. Antibodies used for western blotting: anti-GFP (Mouse monoclonal, Abmart, M20004M), 1:5000; anti-ACTIN (Rabbit

Monoclonal, Beyotime, AF5003), 1:4000. Secondary antibodies: HRP-labeled goat anti-rabbit IgG (H + L) (Abcam, ab205718), 1:15000; HRP-labeled goat anti-mouse IgG (H + L) (Beyotime, A0216), 1:1000.

## Microscopy and analysis

To image larval and adult stages, animals were immobilized in ddH$_2$O with 0.5 M sodium azide and mounted on glass slides before imaging. To image embryos and germ cells, worms were dissected in 2 μl of 0.4× M9 buffer with 0.1 M sodium azide on a coverslip and then mounted on freshly made 1.2–1.4% agarose pads.

To acquire the images shown in Figs. 2d, 3a, 5 and Supplementary Figs. 1, 2, 5g–j, 6a, 6c, 9d, 11 and 12b, a Leica upright DM4 B microscope equipped with a Leica DFC7000 T camera and an HC PL APO 100×/ 1.40-0.70 oil objective was used. Images were taken and processed using Leica Application Suite X software (version 3.7.2.22383) and were rotated and cropped using Adobe Photoshop CS6 software. For the same proteins under different genetic backgrounds, equally normalized images were exported, and contrasts of images were equally adjusted between control and experimental sets. To acquire the images shown in Supplementary Figs. 4b and 5c–f, a Leica upright DM4 B microscope equipped with a Leica DFC7000 T camera and an HC PL FLUOTAR 40×/0.80 objective was used. For images shown in other figures, the Leica THUNDER Imaging System was used, equipped with a K5 sCMOS microscope camera and an HC PL APO 100×/1.40-0.70 oil objective. Images were taken and deconvoluted using Leica Application Suite X software (version 3.7.4.23463). As the intensities of germ granule compartments intensively vary along the adult germline, the display values of fluorescence images showing the relative position of germ granule compartments with E granule components were manually adjusted to visualize these proteins in different germline regions using Leica Application Suite X software (version 3.7.4.23463).

For quantification of the GFP intensity in Fig. 2d, the average fluorescence intensities of 15 worms were analyzed using ImageJ v.1.8.0. For quantification of the P-granule sizes in Supplementary Fig. 12a, 36 granules from the pachytene regions of three animals (3 cells per animal, 4 granules per cell) were picked, and the sizes were calculated using ImageJ v.1.8.0. For quantitative colocalization between different fluorescently labeled proteins, Pearson's R values for the degree of colocalization between two channels in the region defined by the ROI mask were calculated by the Coloc2 plugin in ImageJ2 v.2.3.0. Region of interest (ROI) masks covering individual germ cells were generated using the ROI Manager plugin. At least 15 germ cells in total were selected from 3 independent animals.

## RNA isolation and sequencing

For small RNA deep sequencing, synchronized young adult worms grown at 20 °C were collected. Note, as the egc-1(-);mut-16(-) and elli-1(-);mut-16 mutants exhibited a very high incidence of males (approximately 20% of offspring were males), the males on the cultured plates were manually removed before the worms were collected. Briefly, the animals were sonicated in sonication buffer (20 mM Tris-HCl [pH 7.5], 200 mM NaCl, 2.5 mM MgCl$_2$, and 0.5% NP-40); the eluates were incubated with TRIzol reagent (Invitrogen) prior to isopropanol precipitation. The RNA solution was subjected to DNase I digestion (Thermo Fisher) and re-extracted with TRIzol prior to isopropanol precipitation. Then, 5 μg of total RNA was treated with the RNA processing enzyme RNA 5′-polyphosphatase (Epicentre) to convert 5′-triphosphate RNA or 5′-diphosphorylated RNA to 5′-monophosphate RNA without dephosphorylating monophosphorylated RNA. The RNA solution was re-extracted with TRIzol prior to isopropanol precipitation before library construction.

Small RNAs were subjected to deep sequencing using an Illumina platform (Novogene Bioinformatic Technology Co., Ltd.). Briefly, small RNAs ranging from 17 to 30 nt were gel purified and ligated to a 3′ adaptor (5′-pUCGUAUGCCGUCUUCUGCUUGidT-3′; p, phosphate; idT,

inverted deoxythymidine) and a 5′ adaptor (5′-GUUCAGAGUUCUACA GUCCGACGAUC-3′). The ligation products were gel purified, reverse transcribed, and amplified using Illumina's sRNA primer set (5′-CAA GCAGAAGACGGCATACGA-3′; 5′-AATGATACGGCGACCACCGA-3′). The samples were then sequenced using the Illumina HiSeq platform.

## Small RNA-seq analysis

Illumina-generated raw reads were first filtered to remove adaptors, low-quality tags and contaminants to obtain clean reads. Clean reads ranging from 17 to 30 nt were mapped to the transcriptome assembly WS243 using Bowtie2 v.2.2.5 with default settings. The numbers of reads targeting each transcript were determined using custom Perl scripts. The number of total reads mapped to the genome minus the number of total reads corresponding to sense rRNA transcripts (5S, 5.8S, 18S, and 26S) was used as the normalization value to exclude possible degradation fragments of sense rRNAs. Because some 21U-RNAs and miRNAs overlap with protein-coding genes, reads derived from all known miRNA loci and 21U-RNAs were filtered out prior to comparative analysis. Germline-enriched genes with at least 10 RPM 22G RNAs in wild-type animals were included in the analysis. Note that soma-specific siRNAs were excluded[40]. A cutoff criterion of a twofold change was applied to identify the differentially expressed small RNAs. Scatter plots and Venn diagrams were generated using custom R or Python scripts and modified in Adobe Illustrator. 22G RNA reads were aligned to the C. elegans genome WBcel235 via Bowtie2 v.2.2.5 with default parameters, and IGV v.2.5.3 was used to visualize the alignment results.

All scripts are available upon request.

## Metagene analysis

The metagene profiles were generated according to a method described previously[77]. Bigwig files were generated using a Snakemake workflow (https://gitlab.pasteur.fr/bli/bioinfo_utils). Briefly, the 3′ adapters and 5′ adapters were trimmed from the raw reads using Cutadapt v.2.10 with the following parameters: -a AATGATACGG CGACCACCGA -g CAAGCAGAAGACGGCATACGA –discard-untrimmed. After adapter trimming, the reads containing 18 to 26 nt were selected using bioawk. The selected 18–26 nucleotide reads were aligned to the C. elegans genome (ce11, C. elegans Sequencing Consortium WBcel235) using Bowtie2 v.2.2.5 with the following parameters: -L 6 -i S,1,0.8 -N 0. The resulting alignments were used to generate bigwig files with a custom bash script using BEDtools v.2.30.0, BEDOPS v.2.4.26, and bedGraphToBigWig v.4. Read counts in the bigwig file were normalized to million "nonstructural" mappers— that is, reads containing 18 to 26 nt and mapping to annotations not belonging to "structural" (tRNA, snRNA, snoRNA, rRNA, ncRNA) categories—and counted using featureCounts v.1.6.0. These bigwig files were used to generate "metaprofiles" files with a shell script.

## mRNA-seq analysis

Illumina-generated raw reads were first filtered to remove adaptors, low-quality tags and contaminants to obtain clean reads. Clean reads were mapped to the C. elegans WBcel235 genome using HISAT2 v.2.1.0 with default parameters. Then, the reads were counted via HTSeq-count v.2.0.3 with the following parameters: "-f sam -r name -s no -a 10 -t exon -i gene_id". Differential expression analysis was performed using custom R scripts. A cutoff criterion of a 2-fold change was applied when filtering for differentially expressed genes. All plots were generated using custom R scripts. All scripts are available upon request.

## Quantitative RT–PCR

Total RNA was isolated by Dounce homogenization from the indicated animals and subjected to DNase I digestion (Thermo Fisher). cDNA was synthesized using a HiScript III RT SuperMix Kit (R323-01, Vazyme)

according to the vendor's protocol. qPCR was performed on a Light-Cycler® 480 Real-Time PCR System (Roche) with AceQ qPCR SYBR Green Master Mix (Q111-02, Vazyme). *eft-3* mRNA was used as the internal control for sample normalization. Average Ct values were calculated for three biological replicates with 3 technical replicates of PCR performed in parallel. Relative RNA levels were calculated using the 2-ΔΔCT method. The primers used for RT–qPCR are listed in Supplementary data 4.

### Statistics

Data on bar graphs are presented as the mean values with error bars indicating the SD values. All of the experiments were conducted with independent *C. elegans* animals or the indicated number of replicates (N). Statistical analysis was performed with the two-tailed Student's *t* test or one-way ANOVA with Dunnett's multiple comparison test as indicated. GraphPad Prism 9 or R scripts were used for statistical analysis.

### Reporting summary

Further information on research design is available in the Nature Portfolio Reporting Summary linked to this article.

## Data availability

The raw sequence data reported in this paper have been deposited in the Genome Sequence Archive in the National Genomics Data Center (China National Center for Bioinformation / Beijing Institute of Genomics, Chinese Academy of Sciences) under accession codes CRA013661 and CRA013663. The mass spectrometry proteomics data reported in this paper have been deposited in the iProX repository under accession codes IPX0009095000 and PXD053319. Source data are provided with this paper.

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

## Acknowledgements
We are grateful to the members of the Guang lab for their comments. We are grateful to Dr. Meetali Singh, Dr. Blaise Li and Dr. Germano Cecere for supplying scripts and guiding the Snakemake workflow. We are grateful to Dr. Ruining Cai for help with the bioinformatic analysis. We are grateful to Dr. Gang Wan and Dr. Donglei Zhang for sharing strains. We are grateful to the International *C. elegans* Gene Knockout Consortium and the National Bioresource Project for providing the strains. Some strains were provided by the CGC, which is funded by the NIH Office of Research Infrastructure Programs (P40 OD010440). This work was supported by grants from the National Key R&D Program of China (2022YFA1302700 to S.G., and 2019YFA0802600 awarded to X.F.) and the National Natural Science Foundation of China (32230016 awarded to S.G., 32270583 awarded to C. Z., 32070619 awarded to X.F., 2023M733425 awarded to X.H., and 32300438 awarded to X.H.), and the Strategic Priority Research Program of the Chinese Academy of Sciences (XDB39010600 awarded to S.G.), the Research Funds of Center for Advanced Interdisciplinary Science and Biomedicine of IHM (QYPY20230021 awarded to S.G.) and the Fundamental Research Funds for the Central Universities awarded to S.G.

## Author contributions
S.K. and S.G. conceptualized the research; X.F., Y.S., S.K. and S.G. designed the research; X.C., K.W., F.M. and D.X. performed the research; X.C., K.W., D.X., Chengming Zhu, Xinya Huang, Chenming Zeng, Q.J. and Xiaona Huang contributed new reagents; Y.Y. and M.D. contributed mass spectrometry analysis; K.W. and D.X. contributed analytic tools and performed bioinformatics analysis; S.K. and S.G. wrote the paper.

## Competing interests
The authors declare no competing interests.
