## [Peer Review File · Nature Communications]

REVIEWER COMMENTS

Reviewer #1 (Remarks to the Author):

Assessment: In this submission, Chen et al. looked more closely at the sub-germ-granule partitioning of the RDRP EGO-1, which they found offset from known germ-granule compartments. The researchers utilized EGO-1 IP-MS to identify components of the EGO-1 complex, which found known interactors as well as two proteins with intrinsically disordered regions (IDRs), EGC-1 and ELLI-1. The components of this complex play a crucial role in exogenous RNA interference (RNAi) and collectively reside in what the authors refer to as the "E compartment." This compartment is situated adjacent to the previously defined P, Z, M, and S compartments. A noteworthy observation is that the partitioning of EGO-1/DRH-3/EKL-1 can shift between the M and E compartments during development. On the other hand, the IDR proteins EGC-1 and ELLI-1 remain confined to the E compartment.

The research extends beyond merely identifying a novel sub-compartment within germ granules. It elucidates the interdependence and autonomy in the assembly of P, Z, M, S, and E granules. The study profiles 22G RNA levels and mRNA expression in response to mutations in components of the E compartment. It establishes a subset of E-class genes, which significantly overlap with CSR-1 and EGO-1 germline targets but are distinct from M-class genes. Furthermore, the study demonstrates that EGC-1 and ELLI-1 collaborate in orchestrating the specialized synthesis of 5' small interfering RNAs (siRNAs).

The introduction and overall writing of the paper are excellent. The experiments conducted are thorough with the proper controls. The conclusions drawn are adequately substantiated by the data. Importantly, the discoveries presented in this study represent substantial progress, providing much-needed clarity on the intricate aspects of germ-granule function, siRNA synthesis, the phenomena of sub-granule partitioning, and the role of phase separation in germ cells. Minor and addressable critiques are provided below.

1) IP-MS data to accompany the list provided in Fig 2a should be included in a supplement and deposited in an accessible archive so it can be evaluated.

2) Several of my notes pertain to the proper acknowledgment of the 2017 ELLI-1 PLoS Genetics paper by Andralojc et al., which are admittedly petty things to bring up considering the new findings presented in this submission. But context should be considered. For instance, the authors of this study refer to the 2017 work as recent (line 290), even though it was conducted seven years ago when there were fewer available reagents and limited resolution, and before the Z- and S-

compartments had been described. While recognizing that fixation and immunostaining may introduce differences, it's important to note that the lines generated in this study by Chen et al. also depict a significant amount of ELLI-1 in the rachis/cytoplasm (Fig 3a, S5d, S6a, and S12b), showing that ELLI-1 is not exclusively localized to germ granules. A sweeping statement like "our subcellular localization of ELLI-1 was not consistent with..." feels disingenuous. In addition to *elli-1*'s RNAi and large germ-granule *csr-1/ego-1/drh-3*-like phenotypes and showing the germ-granule localization of ELLI-1, the 2017 study was the first to report:

- 1) The reduced broods of *elli-1* mutants
- 2) Late embryonic expression of ELLI-1 in PGCs (in contrast to none observed in S5h)
- 3) That ELLI-1 contains a low-complexity domain
- 4) The misexpression of RNAi-related genes via microarray – specifically noting *rde-11*
- 5) That ELLI-1 "foci can be found docked next to P granules."

It's noteworthy that the RNAi defects in the *elli-1* mutants generated for this study are much more pronounced than what was reported in the 2017 study. From this, one could argue that the *elli-1* EMS-generated alleles in the 2017 study are hypomorphic, challenging the assumption by Andralojc et al. that they "cause a complete loss-of-function."

3) In the paragraph beginning at line 281, there is a lack of explicit clarification regarding the strains being imaged. The statement "Animals expressing EGC-1::GFP or ELLI-1::GFP" could pertain to either endogenous tags (*ust354*) or integrated arrays (*ustIs272*) utilizing germline promoters and distinct 3'UTRs. I presume that each of these lines demonstrates varying levels of expression or other distinctions. It is crucial to provide absolute clarity on which specific lines are under observation in both the main text and the figure legends.

4) The figure legends should include explanations for differences in exposure levels. Instances of background signal and its absence, such as in the comparison between S6a and S6b, need clarification. Currently, one must refer to the methods section to understand that S6a was captured on a DM4 B and S6b on a Thunder with computational clearing. Including this information directly in the figure legend would enhance accessibility.

5) The developmental distinctions in EGO module colocalization, as highlighted in the paragraph beginning at line 344, were perceptive and intriguing. It would be valuable to emphasize, and quantify, the observation that when the EGO module factors colocalize with MUT-16, it appears that all MUT foci colocalize, while not all EGO foci exhibit the same pattern. Is this the case? If so, it could have significant implications.

6) In Figure 5a, the hypomorphic *drh-3(ne4253)* allele is utilized, and since it is reported to be temperature-sensitive, it is essential to specify whether this experiment was conducted at the permissive or restrictive temperature.

7) Exploring the possibility of synthetic phenotypes in *egc-1*; *elli-1* double mutants could add valuable insights to the study.

8) To provide readers with a clearer understanding of the observed versus expected outcomes, non-parametric statistics should be included alongside the Venn diagrams. Additionally, for enhanced data visualization, consider employing proportional Venn diagrams.

9) In the model, is there evidence that Z granules contain only E-class 22Gs? Or should the 22Gs here be blue?

Reviewer #2 (Remarks to the Author):

This manuscript is a very clear and well-written description of a new compartment of the *C. elegans* germ granule. The authors identify key proteins for the formation of this new compartment, the E compartment, demonstrate how it is organized within the larger germ granule, and show that the EGO-1 RdRP module is associated. Further, the E compartment is a site of CSR-class 22G-RNA biogenesis and is important for synthesis of 22G-RNAs to germline expressed genes. The results are well-supported, the microscopy is beautiful, and the discussion thorough and thoughtful. This manuscript is going to be an important paper in the *C. elegans* RNAi/germ granule field. I have a few minor concerns described below, but nothing that takes away from the overall strength of this paper, for which I strongly support publication at Nature Communications. Also, I like the nomenclature – referring to P, Z, S, M, E as subcompartments within the larger germ granule and individually as P compartment, Z compartment etc. I think this language will simplify the way we discuss germ granule compartments in the future.

Minor comments –

Fig 1C and 3D, missing X-axis labels (could be something simple like “granule comparison” or could move labels from box to below X-axis)

Fig 2, S4 and elsewhere – I would recommend noting in the figures that C14b1.12 is the same gene as *egc-1* (i.e. C15b1.12/*egc-1* or something similar).

Lines 362-364 – it is stated that the EGO module mainly accumulates in the E compartment, but it would be more accurate to say that the EGO module is distributed between the E and M compartments.

Line 359 – the use of the word “differentiation” seems inaccurate. Change to varies across the stages of germ cell progression or varies across the stages of germ cell development.

- Carolyn Phillips

Reviewer #3 (Remarks to the Author):

In this manuscript, the authors identify a novel germ granule compartment that they call the E compartment. They identify several proteins that localize to the E compartment and show that the E-compartment is not affected by disruption of previously known germ granule compartments. They characterize the localization of the E compartment proteins throughout development and find that *EGC-1* and *ELLI-1* are constitutive components of the E-compartment, whereas *EGO-1*, *DRH-3* and *EKL-1* can localize to both the E compartment and the M compartment at specific stages of germ cell development. They also find a function of the E compartment in the biogenesis/stability of a subset of *CSR-1* bound siRNAs that are mapping to the 5' portion of genes. These findings advance the understanding of the complex organization of *C. elegans* germ granules. The manuscript is generally well written and the conclusions are well supported. A few points to be addressed are:

The authors conclude in figure 5a, that *EGC-1::GFP* and *ELLI-1::GFP* granular localization is severely affected in *elli-1(ust203)* and *egc-1(ust134)* respectively. For *EGC-1::GFP* (and other components of the E compartment), this statement is complemented by a western blot (figure 5b) showing no reduction in total protein level supporting that *ELLI-1* is indeed required for *EGC-1* granular localization. For *ELLI-1::GFP*, the overall signal appears reduced in *egc-1(ust134)*. It could be that the loss of *EGC-1* cause a reduction in *ELLI-1* protein levels, which in turn could affect the phase separating properties of *ELLI-1*. This was already determined by western blot for the other E compartment proteins and should also be shown for *ELLI-1*.

Figure S3c, the cell type shown looks like it could perhaps be an oocyte but it should be specified.

Line 376/377: the provided accuracy of the co-occurrence is meaningless. Best round this of to round values (48%, 21%)

Line 481-485: this is rather convoluted. CSR01 class siRNAs are obvious to address, so simply writing “We next analyzed...” would suffice, and reduce the somewhat convoluted language in this part of the manuscript.

Line 501: ‘pools’. Using the term ‘genes’ would be much clearer. I interpreted ‘pools’ as a group of genes, which turned out to be wrong in this context.

Line 550: ‘biogenesis’. This is not correct. The authors analyze steady-state levels, so cannot distinguish between biogenesis (transcription, splicing etc) and decay. Please rephrase

Line 562-567: This was somewhat confusing. Maybe simply write ‘these 138 genes’ in line 562? Also, the idea taken from these results is that the 3’ siRNAs are made by EGO-1 in the cytosol (also see discussion, line 696). This may be correct, but equally likely, in wild-type conditions these are still made in the E compartment. The fact that they are made outside the E compartment in *egc-1* mutants, does not mean they are equally made outside in wild-type setting. I could imagine a scenario where the E compartment enhances the processivity of EGO-1, but that everything happens in E in wild-type animals. Please rephrase these parts to reflect such other options.

In quite some cases (for instance line 680) past tense is used (‘speculated’) where present tense would be the better option.

Typo in line 839: ‘THUNDER’

REVIEWER COMMENTS

Reviewer #1 (Remarks to the Author):

Assessment: In this submission, Chen et al. looked more closely at the sub-germ-granule partitioning of the RDRP EGO-1, which they found offset from known germ-granule compartments. The researchers utilized EGO-1 IP-MS to identify components of the EGO-1 complex, which found known interactors as well as two proteins with intrinsically disordered regions (IDRs), EGC-1 and ELLI-1. The components of this complex play a crucial role in exogenous RNA interference (RNAi) and collectively reside in what the authors refer to as the "E compartment." This compartment is situated adjacent to the previously defined P, Z, M, and S compartments. A noteworthy observation is that the partitioning of EGO-1/DRH-3/EKL-1 can shift between the M and E compartments during development. On the other hand, the IDR proteins EGC-1 and ELLI-1 remain confined to the E compartment.

The research extends beyond merely identifying a novel sub-compartment within germ granules. It elucidates the interdependence and autonomy in the assembly of P, Z, M, S, and E granules. The study profiles 22G RNA levels and mRNA expression in response to mutations in components of the E compartment. It establishes a subset of E-class genes, which significantly overlap with CSR-1 and EGO-1 germline targets but are distinct from M-class genes. Furthermore, the study demonstrates that EGC-1 and ELLI-1 collaborate in orchestrating the specialized synthesis of 5' small interfering RNAs (siRNAs).

The introduction and overall writing of the paper are excellent. The experiments conducted are thorough with the proper controls. The conclusions drawn are adequately substantiated by the data. Importantly, the discoveries presented in this study represent substantial progress, providing much-needed clarity on the intricate aspects of germ-granule function, siRNA synthesis, the phenomena of sub-granule partitioning, and the role of phase separation in germ cells. Minor and addressable critiques are provided below.

Thank you very much for the positive comments on our work.

1) IP-MS data to accompany the list provided in Fig 2a should be included in a supplement and deposited in an accessible archive so it can be evaluated.

Response: Thanks very much for the suggestion.

We included the list of IP-MS data in the new Table S1.

2) Several of my notes pertain to the proper acknowledgment of the 2017 ELLI-1 PLoS Genetics paper by Andralojc et al., which are admittedly petty things to bring up considering the new findings presented in this submission. But context should be considered. For instance, the authors of this study refer to the 2017 work as recent (line 290), even though it was conducted seven years ago

when there were fewer available reagents and limited resolution, and before the Z- and S-compartments had been described. While recognizing that fixation and immunostaining may introduce differences, it's important to note that the lines generated in this study by Chen et al. also depict a significant amount of ELLI-1 in the rachis/cytoplasm (Fig 3a, S5d, S6a, and S12b), showing that ELLI-1 is not exclusively localized to germ granules. A sweeping statement like "our subcellular localization of ELLI-1 was not consistent with..." feels disingenuous. In addition to *elli-1*'s RNAi and large germ-granule *csr-1/ego-1/drh-3*-like phenotypes and showing the germ-granule localization of ELLI-1, the 2017 study was the first to report:

- 1) The reduced broods of *elli-1* mutants
- 2) Late embryonic expression of ELLI-1 in PGCs (in contrast to none observed in S5h)
- 3) That ELLI-1 contains a low-complexity domain
- 4) The misexpression of RNAi-related genes via microarray – specifically noting *rde-11*
- 5) That ELLI-1 "foci can be found docked next to P granules."

It's noteworthy that the RNAi defects in the *elli-1* mutants generated for this study are much more pronounced than what was reported in the 2017 study. From this, one could argue that the *elli-1* EMS-generated alleles in the 2017 study are hypomorphic, challenging the assumption by Andralojc et al. that they "cause a complete loss-of-function."

Response: Thanks very much for the comments.

The *PLoS Genetics* (2017) paper conducted a fabulous forward genetic screening to search for mutants with abnormal P granules and successfully identified a series of proteins required for regulation of the morphological size of P granules, including ELLI-1. Through a series of genetic, cell biology and molecular biology analyses, the study revealed the expression pattern and subcellular localization of ELLI-1 and its functions in modulating RNAi activity, P granule accumulation, fertility and gene expression in the germline. These findings, together with our results, revealed the existence of E granules and their important biological functions in germ cell development. We are very sorry for missing these important findings in the previous draft.

We have revised our manuscript to include these outstanding discoveries and also emphasized that E granules also present in the rachis/cytoplasm of *C. elegans* germline.

Line 196-199: "We observed that GFP::EGO-1 largely accumulated in perinuclear foci, which is consistent with a previous report ¹, and also formed a considerable number of visible aggregates in the rachis of the germline (Figs. 1a, S2b-S2d)".

Line 247-248: "ELLI-1 was previously identified to function with CSR-1 to modulate RNAi activity, P granule morphology, fertility and gene expression in the germline ²".

Line 257-261: "It is noteworthy that the RNAi defects in the *elli-1* mutants generated for this study are much more pronounced than previously reported ². We

speculate that the previous EMS mutagenesis-generated *elli-1* alleles are likely hypomorphic, which may not be null alleles and cause partial loss-of-function.”

Line 263-264: “Similar to *ego-1* mutants and a previous report ², animals lacking EGC-1 or ELLI-1 exhibited fertility defects;”

Line 275-276: “Indeed, the *elli-1* mRNA has been reported to enrich in the germline ².”

Line 291-294: “We found that these proteins mainly accumulated on the surface of the adult germline, likely in perinuclear foci surrounding germ cell nuclei, and formed a considerable amount of smaller aggregates in the rachis (Figs. 3a and S5j), which was consistent with the subcellular localization pattern of GFP::EGO-1 (Fig. S2d), and ELLI-1::GFP ².”

Line 301-304: “Additionally, simultaneous imaging of ELLI-1::GFP(*ust374*, *in situ*) and mCherry::CGH-1 both on the surface and in the rachis of the germline revealed that ELLI-1::GFP did not colocalize with mCherry::CGH-1, suggesting that ELLI-1 does unlikely accumulate in the P-body (Fig. S6b).”

Line 309-313: “ELLI-1::GFP was barely detectable in early embryos, implying that the expression of ELLI-1 may be inhibited during early embryo development (Fig. S5h). Consistently, the zygotic ELLI-1 was shown to begin to accumulate in the cytoplasm of primordial germ cells between the comma to 2-fold stage of embryogenesis ².”

Line 317-319: “However, in early embryos, only GFP and GFP::HIS-58, but not ELLI-1::GFP and tagRFP::ELLI-1 could be detected, suggesting a possible posttranscriptional regulation of *elli-1* gene in early embryos (Fig. S6c).”

Line 335-336: “Consistently, perinuclear ELLI-1 foci were shown to dock next to P granules in germ cells ².”

Line 532-533: “Furthermore, several RNAi-related genes, including the *rde-11* gene, were misexpressed upon a hypomorphic mutation in the *elli-1* gene ².”

3) In the paragraph beginning at line 281, there is a lack of explicit clarification regarding the strains being imaged. The statement "Animals expressing EGC-1::GFP or ELLI-1::GFP" could pertain to either endogenous tags (*ust354*) or integrated arrays (*ustIs272*) utilizing germline promoters and distinct 3'UTRs. I presume that each of these lines demonstrates varying levels of expression or other distinctions. It is crucial to provide absolute clarity on which specific lines are under observation in both the main text and the figure legends.

Response: Thanks very much for the suggestion.

We have revised the manuscript to provide detailed information of the strains in the main text and legends of Figs. S6a and S6b.

4) The figure legends should include explanations for differences in exposure levels. Instances of background signal and its absence, such as in the comparison between S6a and S6b, need clarification. Currently, one must refer to the methods section to understand that S6a was captured on a DM4 B and S6b on a Thunder with computational clearing. Including this information directly in the figure legend would enhance accessibility.

Response: Thanks very much for the comment.

We have revised the manuscript to provide detailed information of the methods of image acquisition and processing in the legends of Figs. 1, 3, 4, 5, S6, S8, S9 and S12.

5) The developmental distinctions in EGO module colocalization, as highlighted in the paragraph beginning at line 344, were perceptive and intriguing. It would be valuable to emphasize, and quantify, the observation that when the EGO module factors colocalize with MUT-16, it appears that all MUT foci colocalize, while not all EGO foci exhibit the same pattern. Is this the case? If so, it could have significant implications.

Response: Thanks very much for bringing up this concern.

We completely agree with you that germ cells may finely control the distribution of EGO module between E granules and Mutator foci.

The EGO module constitutively localized in E granules, yet the distribution of the EGO module in Mutator foci varies across the stages of germ cell development. We quantified the percentage of Mutator foci with EGO module components (new Fig. S9c) and found that the majority of Mutator foci enclosed the EGO module in mitotic and transition zone, and late pachytene.

We add this information in the main text.

Line 357-362: “The EGO module factors constitutively colocalized with markers of E granules (tagRFP::ELLI-1) throughout the germline and partially colocalized with Mutator foci (mCherry::MUT-16) in the mitotic, transition and late pachytene regions of the germline (Figs. S9a-S9c). In the early/mid pachytene region of the adult germline, the EGO module factors colocalized with components of E granules but rarely with Mutator foci (Fig. S9a-S9c).”

As you suggested, we are still trying hard to use the Turbo-ID method to identify other components in E granules and Mutator foci. Hopefully, we could answer your questions in the near future.

6) In Figure 5a, the hypomorphic *drh-3(ne4253)* allele is utilized, and since it is reported to be temperature-sensitive, it is essential to specify whether this experiment was conducted at the permissive or restrictive temperature.

Response: Thanks very much for the suggestion.

The *drh-3(ne4253)* lesion (T834M) alters a residue that contacts RNA in the Vasa crystal structure³. Although the *drh-3(ne4253)* point mutants exhibit a range of phenotypes that increase in penetrance at 25 °C, in terms of brood size and the Him phenotype, they do not appear to be classic temperature-sensitive mutants³. The hypomorphic *drh-3(ne4253)* allele was reported to cause deficient RNAi response, siRNA biogenesis defects and dramatically reduced brood size at 20 °C³. Thus, we conducted the experiment at 20 °C.

We have revised the figure legend to include the temperature information.

Line 1261-1265: “The *drh-3(ne4253)* lesion (T834M) alters a residue that contacts RNA in the Vasa crystal structure³. The hypomorphic *drh-3(ne4253)* allele was reported to cause RNAi defective phenotype, siRNA biogenesis defects and dramatically reduced brood size at 20 °C³. The indicated animals were grown at 20 °C.”

7) Exploring the possibility of synthetic phenotypes in *egc-1; elli-1* double mutants could add valuable insights to the study.

Response: Thanks very much for the suggestion.

We have generated *egc-1(ust134);elli-1(ust203)* and *egc-1(ust206);elli-1(ust203)* animals. The brood size of these animals was counted (new Fig. S4c) and did not reveal further reduction of brood size, compared with that of *elli-1(ust203)* animals.

We add this information in the main text.

Line 268-271: “Animals lacking both EGC-1 and ELLI-1 did not show further reduction of brood size, compared with that of *elli-1(-)* animals, suggesting that EGC-1 and ELLI-1 may regulate reproduction in the same pathway (Fig. S4c).”

8) To provide readers with a clearer understanding of the observed versus expected outcomes, non-

parametric statistics should be included alongside the Venn diagrams. Additionally, for enhanced data visualization, consider employing proportional Venn diagrams.

Response: Thanks very much for the suggestion.

We revised the figures as you suggested, employed proportional Venn diagrams and included non-parametric statistics alongside the Venn diagrams in Figs. 6c-6e, 7c, S13b, S13c, S14a, S14b, S15b and S16c. Additionally, we rechecked our bioinformatic analysis data and found that one gene (Y105E8A.1) was mistakenly omitted in our previous analysis. We corrected the data analysis accordingly

9) *In the model, is there evidence that Z granules contain only E-class 22Gs? Or should the 22Gs here be blue?*

Response: Thanks very much for bringing up this concern.

WAGO-4 binds to Mutator foci-derived siRNAs upon feeding RNAi treatment and promotes transgenerational inheritance ⁴. Additionally, the WAGO-4 targets mainly overlaps with those of CSR-1 and partially overlaps with those of WAGO-1 ⁴. These results together suggested that WAGO-4 could bind to siRNAs generated in both E granules and Mutator foci. Therefore, Z-granules likely contain both E- and M-class 22G RNAs.

We revised the model and the figure legend to include this information.

Line 1335-1342: “E granules-derived 22G RNAs are bound to CSR-1, while 22G RNAs derived from Mutator foci are bound to WAGO-1 and HRDE-1. WAGO-4 binds to Mutator foci-derived siRNAs upon feeding RNAi treatment and promotes transgenerational inheritance ⁴. Additionally, the endogenous WAGO-4 targets mainly overlap with those of CSR-1 and partially overlap with those of WAGO-1 ⁴, suggesting that WAGO-4 could bind to siRNAs generated in both E granules and Mutator foci and Z granules likely contain both E- and M-class 22G RNAs.”

Reviewer #2 (Remarks to the Author):

This manuscript is a very clear and well-written description of a new compartment of the C. elegans germ granule. The authors identify key proteins for the formation of this new compartment, the E compartment, demonstrate how it is organized within the larger germ granule, and show that the EGO-1 RdRP module is associated. Further, the E compartment is a site of CSR-class 22G-RNA biogenesis and is important for synthesis of 22G-RNAs to germline expressed genes. The results are well-supported, the microscopy is beautiful, and the discussion thorough and thoughtful. This manuscript is going to be an important paper in the C. elegans RNAi/germ granule field. I have a

few minor concerns described below, but nothing that takes away from the overall strength of this paper, for which I strongly support publication at Nature Communications. Also, I like the nomenclature – referring to P, Z, S, M, E as subcompartments within the larger germ granule and individually as P compartment, Z compartment etc. I think this language will simplify the way we discuss germ granule compartments in the future.

Thank you very much for the positive comments on our work.

Minor comments –

Fig 1C and 3D, missing X-axis labels (could be something simple like “granule comparison” or could move labels from box to below X-axis)

Response: Thanks very much for the comment.

We have revised the X-axis labels in Figs. 1c and 3d as suggested.

*Fig 2, S4 and elsewhere – I would recommend noting in the figures that C14b1.12 is the same gene as *egc-1* (i.e. C15b1.12/*egc-1* or something similar).*

Response: Thanks very much for the comment.

We have revised Figs. 2 and S4 and included the information about of the *egc-1* gene.

Lines 362-364 – it is stated that the EGO module mainly accumulates in the E compartment, but it would be more accurate to say that the EGO module is distributed between the E and M compartments.

Response: Thanks very much for the comment.

We have revised the main text as suggested.

Line 369-372: “We conclude that EGC-1 and ELLI-1 localize to a new subcompartment of the germ granule, which we name the E granule, and the EGO module is distributed between E granules and Mutator foci.”

Line 359 – the use of the word “differentiation” seems inaccurate. Change to varies across the stages of germ cell progression or varies across the stages of germ cell development.

Response: Thanks very much for the suggestion.

We have revised the main text as suggested.

Line 364-367: “These data suggest that the EGO module could localize to both the E and M compartments of the germ granule and the relative distribution of the EGO module to these compartments varies across the stages of germ cell development.”

- Carolyn Phillips

Reviewer #3 (Remarks to the Author):

In this manuscript, the authors identify a novel germ granule compartment that they call the E compartment. They identify several proteins that localize to the E compartment and show that the E-compartment is not affected by disruption of previously known germ granule compartments. They characterize the localization of the E compartment proteins throughout development and find that EGC-1 and ELLI-1 are constitutive components of the E-compartment, whereas EGO -1, DRH-3 and EKL-1 can localize to both the E compartment and the M compartment at specific stages of germ cell development. They also find a function of the E compartment in the biogenesis/stability of a subset of CSR-1 bound siRNAs that are mapping to the 5' portion of genes. These findings advance the understanding of the complex organization of C. elegans germ granules. The manuscript is generally well written and the conclusions are well supported. A few points to be addressed are:

Thank you very much for the positive comments on our work.

The authors conclude in figure 5a, that EGC-1::GFP and ELLI-1::GFP granular localization is severely affected in elli-1(ust203) and egc-1(ust134) respectively. For EGC-1::GFP (and other components of the E compartment), this statement is complemented by a western blot (figure 5b) showing no reduction in total protein level supporting that ELLI-1 is indeed required for EGC-1 granular localization. For ELLI-1::GFP, the overall signal appears reduced in egc-1(ust134). It could be that the loss of EGC-1 cause a reduction in ELLI-1 protein levels, which in turn could affect the phase separating properties of ELLI-1. This was already determined by western blot for the other E compartment proteins and should also be shown for ELLI-1.

Response: Thanks very much for the comments.

We have conducted a new western blotting assay to detect the protein level of ELLI-1 in different genetic backgrounds (new Fig. 5b). The protein level of ELLI-1 is dramatically reduced in *egc-1(ust134)* mutants, compared to that in wild-type animals.

We have added this information in the main text.

Line 405-407: “The protein level of ELLI-1 was dramatically reduced in *egc-1(-)* mutants, suggesting that the loss of EGC-1 causes a reduction of ELLI-1 proteins,

which might in turn affect the formation of ELLI-1 foci (Fig. 5b).”

Figure S3c, the cell type shown looks like it could perhaps be an oocyte but it should be specified.

Response: Thanks very much for the comment.

We have included the cell type information in the legend of Fig. S3c.

Line 376/377: the provided accuracy of the co-occurrence is meaningless. Best round this of to round values (48%, 21%)

Response: Thanks very much for the comment.

We agree with the reviewer and round the numbers to round values (**lines 381-385**).

Line 481-485: this is rather convoluted. CSR01 class siRNAs are obvious to address, so simply writing “We next analyzed...” would suffice, and reduce the somewhat convoluted language in this part of the manuscript.

Response: Thanks very much for the comment.

We have revised the text as suggested.

Line 487-493: “Current models posit that CSR-1 class siRNAs are synthesized by the RdRP EGO-1 and that WAGO class siRNAs are likely synthesized by the Mutator foci-localized RdRP RRF-1. Because EGO-1 localizes to E granules, we tested whether the E-class siRNAs were likely to be CSR-1-class siRNAs. Indeed, a comparison of our list of E-class siRNAs with the published lists of CSR-1 bound siRNAs showed that the E-class siRNAs represented a subset of the CSR-1-class siRNAs (Fig. S14a).”

Line 501: ‘pools’. Using the term ‘genes’ would be much clearer. I interpreted ‘pools’ as a group of genes, which turned out to be wrong in this context.

Response: Thanks very much for the comment.

We have revised the text as suggested (**Line 505**).

Line 550: ‘biogenesis’. This is not correct. The authors analyze steady-state levels, so cannot

distinguish between biogenesis (transcription, splicing etc) and decay. Please rephrase

Response: Thanks very much for the comment.

We have rephrased the sentence.

Line 553-555: “For example, the accumulation of *klp-7* and *cls-2* mRNAs was unchanged in both *egc-1* and *elli-1* mutants (Fig. S16b).”

*Line 562-567: This was somewhat confusing. Maybe simply write ‘these 138 genes’ in line 562? Also, the idea taken from these results is that the 3’ siRNAs are made by EGO-1 in the cytosol (also see discussion, line 696). This may be correct, but equally likely, in wild-type conditions these are still made in the E compartment. The fact that they are made outside the E compartment in *egc-1* mutants, does not mean they are equally made outside in wild-type setting. I could imagine a scenario where the E compartment enhances the processivity of EGO-1, but that everything happens in E in wild-type animals. Please rephrase these parts to reflect such other options.*

Response: Thanks very much for the suggestion.

We totally agree with the idea that the fact that the 3’ siRNAs are made outside the E granule in *egc-1* mutants does not necessarily mean they are equally made outside in wild-type animals. As you suggested, we are still trying hard to use the Turbo-ID method to identify other components in E granules and Mutator foci. The identification of new components and regulators will hopefully help us answer your questions in the near future.

We have revised the manuscript to include your suggestions of E granule’s function on 3’ siRNAs production.

Line 567-571: “We tested a number of E class genes by qRT–PCR and found that the expression levels of E-class genes depend on EGO-1, but not EGC-1 and ELLI-1, hinting their different degrees of necessity in promoting the generation of E class siRNAs, although E class siRNAs were all significantly reduced in *egc-1*, *elli-1* and *ego-1* mutants (Fig. S16f).”

Line 705-709: “Alternatively, both 5’ siRNAs and 3’ siRNAs may be generated in E granules in wild-type animals and cells may take unknown measures to sustain 3’ siRNAs production in the cytosol upon defective E granule assembly. The fact that the 3’ siRNAs are made outside E granules in *egc-1* or *elli-1* mutants does not necessarily mean they are equally made outside in wild-type animals. Further work is needed to understand how and why the EGO module might produce both 5’ and 3’ CSR-1 class siRNAs and whether and how they might do so in distinct regions of the cell.”

In quite some cases (for instance line 680) past tense is used ('speculated') where present tense would be the better option.

Response: Thanks very much for the suggestion.

We have carefully checked the manuscript and revised the tense of statement.

Typo in line 839: 'THUNDER'

Response: corrected.

References

1. Claycomb JM, *et al.* The Argonaute CSR-1 and Its 22G-RNA Cofactors Are Required for Holocentric Chromosome Segregation. *Cell* **139**, 123-134 (2009).
2. Andralojc KM, *et al.* ELLI-1, a novel germline protein, modulates RNAi activity and P-granule accumulation in *Caenorhabditis elegans*. *Plos Genet* **13**, (2017).
3. Gu W, *et al.* Distinct argonaute-mediated 22G-RNA pathways direct genome surveillance in the *C. elegans* germline. *Mol Cell* **36**, 231-244 (2009).
4. Xu F, *et al.* A Cytoplasmic Argonaute Protein Promotes the Inheritance of RNAi. *Cell Rep* **23**, 2482-2494 (2018).

REVIEWERS' COMMENTS

Reviewer #1 (Remarks to the Author):

All of my comments were addressed.

Reviewer #2 (Remarks to the Author):

Overall, the authors have addressed the comments made by all reviewers. I have only a few minor points that mostly stem from updated text in the revision.

Reviewer 1 noted that the mass spec data should be made accessible in a publicly accessible archive. That does not seem to have been done yet, and should be prior to publication.

In response to another suggestion by reviewer 1, text was added indicating that WAGO-4 binds mutator foci-derived siRNAs upon RNAi treatment (line 1335). While there is data indicating that WAGO-4 binds small RNAs derived from the RNAi treatment, as far as I know, it has not been shown that these siRNAs are Mutator-derived. Unless it has been shown that WAGO-4 does not bind these small RNAs in a mutator pathway mutant (maybe it has?), I would argue that it is also possible that EGO-1 is producing secondary siRNAs from the RNAi treated gene. Unless it has been clearly shown that these secondary siRNAs are derived from exogenous RNAi are dependent on the mutator proteins, I would suggest editing the text to simply say that it is unknown whether these siRNAs are produced by the Mutator complex or EGO-1.

Line 304 – awkward text, I would suggest something like “... ELLI-1::GFP does not colocalize with mCherry CGH-1, indicating that ELLI-1 does not accumulate in the P-body.”

Line 505 – I would suggest something like “siRNA-seq analysis identified 199 genes for which the mapped siRNAs increased in abundance in *egc-1* or *elli-1* mutants.”

Line 698-699 – I would suggest “siRNAs mapping to the 3' regions of CSR-1 class mRNAs may be produced by EGO-1 in the cytosol.”

Reviewer #3's concern about the confusing nature of the term siRNA pool is not full addressed as the term is used throughout lines 458-481 and in Materials and Methods. I would recommend changing all of these for clarity.

Reviewer #3 (Remarks to the Author):

The authors addressed the comments well and I have no further issues. I recommend acceptance of this manuscript

REVIEWERS' COMMENTS

Reviewer #1 (Remarks to the Author):

All of my comments were addressed.

Thank you very much for the positive comments on our work.

Reviewer #2 (Remarks to the Author):

Overall, the authors have addressed the comments made by all reviewers. I have only a few minor points that mostly stem from updated text in the revision.

Thank you very much for the positive comments on our work.

Reviewer 1 noted that the mass spec data should be made accessible in a publicly accessible archive. That does not seem to have been done yet, and should be prior to publication.

Response: Thanks very much for the suggestion.

We have deposited the mass spectrometry proteomics data in the OMIX (China National Center for Bioinformation / Beijing Institute of Genomics, Chinese Academy of Sciences) under the accession code OMIX006566 [<https://ngdc.cnbc.ac.cn/omix/release/OMIX006566>] and included the detailed information of IP-MS data in the new Supplementary data 1.

In response to another suggestion by reviewer 1, text was added indicating that WAGO-4 binds mutator foci-derived siRNAs upon RNAi treatment (line 1335). While there is data indicating that WAGO-4 binds small RNAs derived from the RNAi treatment, as far as I know, it has not been shown that these siRNAs are Mutator-derived. Unless it has been shown that WAGO-4 does not bind these small RNAs in a mutator pathway mutant (maybe it has?), I would argue that it is also possible that EGO-1 is producing secondary siRNAs from the RNAi treated gene. Unless it has been clearly shown that these secondary siRNAs are derived from exogenous RNAi are dependent on the mutator proteins, I would suggest editing the text to simply say that it is unknown whether these siRNAs are produced by the Mutator complex or EGO-1.

Response: Thanks very much for bringing up this concern.

We have revised the legend as suggested.

Lines 1365-1368: “WAGO-4 binds to siRNAs derived from the exogenous RNAi

treatment and promotes transgenerational inheritance. It is currently unknown whether these siRNAs are produced by the Mutator complex in Mutator foci or EGO-1 in E granules.”

Line 304 – awkward text, I would suggest something like “... ELLI-1::GFP does not colocalize with mCherry CGH-1, indicating that ELLI-1 does not accumulate in the P-body.”

Response: Thank you very much for the suggestion.

We have revised the manuscript as suggested.

Lines 307-308: “...ELLI-1::GFP did not colocalize with mCherry::CGH-1, indicating that ELLI-1 does not accumulate in the P-body (Supplementary Fig. 6b).”

*Line 505 – I would suggest something like “siRNA-seq analysis identified 199 genes for which the mapped siRNAs increased in abundance in *egc-1* or *elli-1* mutants.”*

Response: Thank you very much for the suggestion.

We have revised the manuscript as suggested.

Lines 513-514: “siRNA-seq analysis identified 199 genes for which the mapped siRNAs increased in abundance in *egc-1* or *elli-1* mutants”.

Line 698-699 – I would suggest “siRNAs mapping to the 3' regions of CSR-1 class mRNAs may be produced by EGO-1 in the cytosol.”

Response: Thanks very much for the suggestion.

We have revised the manuscript as suggested.

Lines 708-709: “...suggesting that the siRNAs mapping to the 3' regions of CSR-1 class mRNAs may be produced by EGO-1 in the cytosol.”.

Reviewer #3's concern about the confusing nature of the term siRNA pool is not full addressed as the term is used throughout lines 458-481 and in Materials and Methods. I would recommend changing all of these for clarity.

Response: Thanks very much for the suggestion.

We have carefully checked this term throughout the manuscript and revised the

manuscript as suggested.

Reviewer #3 (Remarks to the Author):

The authors addressed the comments well and I have no further issues. I recommend acceptance of this manuscript

Thank you very much for the positive comments on our work.